# Observing Low Altitude Features in Ozone Concentrations in a Shoreline Environment via Uncrewed Aircraft Systems

Josie K. Radtke[1],
Benjamin N. Kies[1],
Whitney A. Mottishaw[1],
Sydney M. Zeuli[1],
Aidan T.H. Voon[1],
Kelly L. Koerber[1],
Grant W. Petty[2],
Michael P. Vermeuel[3†],
Timothy H. Bertram[3],
Ankur R. Desai[2],
Joseph P. Hupy[4],
R. Bradley Pierce[2,5],
Timothy J. Wagner[5],
Patricia A. Cleary[1]

[1]Department of Chemistry, University of Wisconsin-Eau Claire, 105 Garfield Ave, Eau Claire, WI, USA

[2]Department of Atmospheric and Oceanic Sciences, University of Wisconsin-Madison, 1225 W Dayton St, Madison, WI, USA

[3]Department of Chemistry, University of Wisconsin-Madison, 500 Lincoln Dr, Madison, WI, USA

[4]Aviation and Transportation Technology, Purdue Polytechnic Institute, Purdue University, West Lafayette, IN, USA

[5]Space Sciences and Engineering Center, University of Wisconsin-Madison, Madison WI, USA

[†]Now at Department of Soil, Water, and Climate, University of Minnesota, St. Paul, MN, USA

*Correspondence to*: Patricia A. Cleary (clearypa@uwec.edu)

**Abstract.** Ozone is a pollutant formed in the atmosphere by photochemical processes involving nitrogen oxides ($NO_x$) and volatile organic compounds (VOCs) when exposed to sunlight. Tropospheric boundary layer ozone is regularly measured at ground stations and sampled infrequently through balloon, lidar, and crewed aircraft platforms, which have demonstrated characteristic patterns with altitude. Here, to better resolve vertical profiles of ozone within the atmospheric boundary layer, we developed and evaluated an uncrewed aircraft system (UAS) platform for measuring ozone and meteorological parameters of temperature, pressure, and humidity. To evaluate this approach, an UAS was flown with a portable ozone monitor and a meteorological temperature and humidity sensor to compare to tall tower measurements in northern Wisconsin. In June 2020, as a part of the WiscoDISCO20 campaign, a DJI M600 hexacopter UAS was flown with the same sensors to measure Lake Michigan shoreline ozone concentrations. This latter UAS experiment revealed low-altitude structure in ozone concentrations in a shoreline environment showing highest ozone at altitudes from 20-100 mAGL. These first such measurements of low-altitude ozone via UAS in the Great Lakes Region revealed a very shallow layer of ozone rich air lying above the surface.

## 1 Introduction

Ozone at elevated concentrations near the surface is a pollutant that causes respiratory irritation in humans (Bell et al., 2006; Brauner et al., 2016 ) and oxidative stress on photosynthesizing organisms in many ecosystems (Fuhrer, 2002). In the troposphere, ozone is generated by reactions of nitrogen oxides ($NO_x = NO + NO_2$) and volatile organic compounds (VOCs) exposed to sunlight (Sillman, 1999). $NO_x$ compounds are emitted from combustion sources and VOCs are emitted by biogenic processes and anthropogenic industrial sources such as transportation and evaporated solvents (benzene, formaldehyde, ethyl alcohol, etc.). While ozone is monitored at the surface to meet various air quality monitoring standards, or to understand ozone depositional losses, ozone gradients aloft have been measured in various ways over the years using sondes that reach the stratosphere (Beekmann et al., 1995; Witte et al., 2018), instrumented towers (Crawford et al., 1996; Desjardins et al., 1995), tethered balloons (Chandrasekar et al., 2003; Li et al., 2018; Mazzuca et al., 2017; Zhang et al., 2019; Tang et al., 2021; Demuer et al., 1997; Greenberg et al., 2009; Knapp et al., 1998), and crewed aircraft (e.g. (Kaser et al., 2017; Crawford et al., 1996; Tanimoto et al., 2015; Tarasick et al., 2019; Desjardins et al., 1995). Because ozone is generated by chemical reactions, the confinement of primary pollutants near the surface via atmospheric inversions tends to produce higher ozone concentration events at the surface. Understanding the volume of air in and above an inversion at a shoreline location prone to high ozone events can help elucidate the chemical evolution processes in this environment (Chai et al., 2013; Tang et al., 2021; Tang et al., 2009).

Recently there have been an expansion of efforts for Uncrewed Aircraft Systems (UAS) to be used for atmospheric profiling (Telg et al., 2017; Chilson et al., 2019; De Boer et al., 2021; Hemingway et al., 2017; Jacob et al., 2018; Koch et al., 2018; Wainwright et al., 2015; Li et al., 2018 ). Tethered balloons have been used to study vertical ozone (Demuer et al., 1997; Peng et al., 2008; Knapp et al., 1998; Zhang et al., 2019; Greenberg et al., 2009), and meteorological conditions (Chandrasekar et

al., 2003) gathering data at heights ranging from ground level to 1500 meters above ground level, which included evaluations of episodes of biomass burning (Xu et al., 2018) and mesoscale modeling of ozone in the upper troposphere (Peng et al., 2008). UAS platforms measuring atmospheric properties have deployed at heights ranging from ground level to 4000 meters above ground level (Adkins and Sescu, 2017; Chilson et al., 2019; Cook et al., 2013; Greatwood et al., 2017; Hemingway et al., 2017). The portable Personal Ozone Monitor (2B Tech POM) mounted on a UAS performed consistently in comparison to a larger ozone photoanalyzer equipped to a tethered airship in the lower troposphere (Li et al., 2018) but with some significant discrepancies between platforms within the planetary boundary layer. Through modeling efforts using Generalized Additive Models (GAMs) Li et al. (2018) attributed these discrepancies to horizontal separation of platforms and vertical variations in atmospheric structure including temperature and relative humidity.

The effect of lake breeze or sea breeze on regional ozone in shoreline environments has been a point of interest in several studies. The association of sea breezes and lake breezes with elevated ozone at shoreline locations have been documented in Houston  (Banta et al., 2005), Toronto  (Levy et al., 2010; Sills et al., 2011), New York City during LISTOS (Zhang et al., 2020) , and near Chesapeake Bay (Gronoff et al., 2019), but few studies have explored vertical profiles within the marine layer structure. The lake and sea breeze meteorology develop from colder air parcels moving over land underneath buoyant warmer air which can create capping inversion which can trap pollutants (Lu and Turco, 1994; Gaza, 1998; Levy et al., 2010; Sills et al., 2011). Multiple groups have found there to be a notable difference in ozone levels during a sea or lake breeze including OWLETS (The Ozone Water-Land Transition Study) in the Chesapeake Bay region (Sullivan et al., 2019), ABLE (Amazon Boundary Layer Experiment) over Manaus Brazil (Guimaras et al., 2020), and a research team in the Salt Lake City region (Blaylock et al., 2017). OWLETS analyzed ozone pollution using ozone sensors mounted onto ships and UAS. These measurements showed that ozone builds up over the bay due to the effect of sea breeze up to 2000 m above sea level (Sullivan et al., 2019). With these observations, Sullivan et al. (2019) attempted to forecast chemical emissions based upon emissions from ships and other emission sources in the bay.  During ABLE, Guimaras et al. (2020) used UAS to study the urban nighttime boundary layer over Manaus, Brazil in both the dry and wet seasons. They conducted flights from the center of the city from ground level up to 500 m to quantify the effect humidity has on ozone pollution over Manaus at night (Guimaras et al., 2020). Crewed aircraft were used over the Great Salt Lake in Utah to study ozone levels up to 4000 m above ground level and demonstrated complicating factor of lake breeze transporting contrasting air masses into the region (Blaylock et al., 2017; Crosman et al., 2017; Horel et al., 2016).

The relationship between ambient ozone and coastal environments has been investigated by aircraft, mobile platforms for the 2017 Lake Michigan Ozone Study (LMOS) (Cleary et al., 2022b; Doak et al., 2021; Stanier et al., 2021)  and UAS for the OWLETs campaign (Gronoff et al., 2019) and multi-UAS strategies for WiscoDISCO-21 (Tirado et al., 2023; Cleary et al., 2022a). Ozone concentrations have been shown to vary with altitude sharply in low-altitude crewed aircraft flights over Lake Michigan (Cleary et al., 2022b; Stanier et al., 2021). During the OWLETs campaign, the high-over-water ozone was investigated by UAS and ship-based platforms, including low ozone titration events. In these transitional environments, model and observation agreement can be improved with the capture of small gradients and modelling marine inversions over water

(Abdi-Oskouei et al., 2020; Mcnider et al., 2018; Cleary et al., 2015). Recent observations over riverine environments have demonstrated the viability of UAS for detecting low altitude variations in ozone and plume chemistry (Li et al., 2021; Guimaras et al., 2020; Ye et al., 2022). The horizontal extent of lake breeze has also been documented at the shoreline to Lake Michigan where horizontal gradients close to the shoreline were observed during 2017 LMOS (Cleary et al., 2022b; Stanier et al., 2021).

The goal of this study is to develop a technique for investigating the vertical profiles of ozone at a shoreline location impacted by high ozone episodes. Chiwaukee Prairie, WI hosts a regulatory site at a shoreline state natural area, which is one of the few in Wisconsin which regularly exceed federal ozone standards and is regularly impacted by lake breeze. The large sources of emissions for ozone precursors are mainly concentrated in the Chicago metro area and the presence of Lake Michigan provides an inverted atmosphere at times in which to trap said pollutants. The role of the inversion over Lake Michigan, the advection of pollutants over Lake Michigan and then back on land during the meso-scale meteorological phenomenon of the Lake Breeze is the focus of the WiscoDISCO field campaigns. We first outline how the instrumentation was tested in a non-lake shore environment during CHEESEHEAD19 and then describe improvements to instrumentation performance for the first WiscoDISCO field campaign in 2020. Here, the UAS-based observations of ozone and meteorological variables were compared to tower observations in a forested environment in 2019 and then ground observations at a Lake Michigan shoreline in 2020 demonstrating improved performance and viability of a UAS atmospheric profiler to investigate lower atmospheric variability at a site impacted by lake breeze and poor air quality.

## 2 Materials and Methods

### 2.1 CHEESEHEAD19 and PEcorINO Measurement Campaigns

The University of Wisconsin-Eau Claire team joined the Chequamegon Heterogenous Ecosystem Energy-Balance Study Enabled by a High-density Extensive Array of Detectors 2019 (CHEESEHEAD19) campaign (Butterworth et al., 2021) in July 2019 in order to compare UAS-based observations with tower observations made during the first 7-day intensive observation period of the field campaign. CHEESEHEAD19 was the multi-institute campaign that sought to give insight into atmosphere-land exchanges in a temperate mixed forest (Butterworth et al., 2021). The CHEESEHEAD19 domain incorporated a swath of land in the Chequamegon National Forest near Park Falls, WI, where multiple tower, UAS, aircraft, ground, and remote sensing observations were conducted, focused around the 447 m instrumented tower operated by WLEF-TV (45.946 N, 90.273 W) and owned by the State of Wisconsin. Local vehicular traffic at the tall tower site was light and mixed trucking, forestry, and automobile traffic on WI Highway 182 (Figure 1). The tower has been in operation since 1995 as a National Oceanic and Atmospheric Administration (NOAA) greenhouse gas tall tower site (LEF) and since 1996 as an Ameriflux eddy covariance site (US-PFa), with sampling inlets and flux measurements currently at 30, 122 and 396 m above ground level. Ozone concentration observations were at two specific heights on the tower (30 m and 122 m) by two instruments: a chemical ionization time-of-flight mass spectrometer (CI-ToFMS, Tofwerk, and Aerodyne) using oxygen anion ($O_2^-$) ionization chemistry (1 s L.O.D. ~10 ppt) (Novak et al., 2020) and an EPA standard photometric analyzer (L.O.D. 0.5

ppb ThermoScientific 49i) (Vermeuel et al., 2021). The fast observations of ozone by the CI-ToFMS instrument were used
for flux measurements (Vermeuel et al., 2021). For the purposes of proving the viability of a UAS-mounted ozone
measurement, the tower ozone measurements were compared to ozone gradient measurements from the UAS-mounted POM.

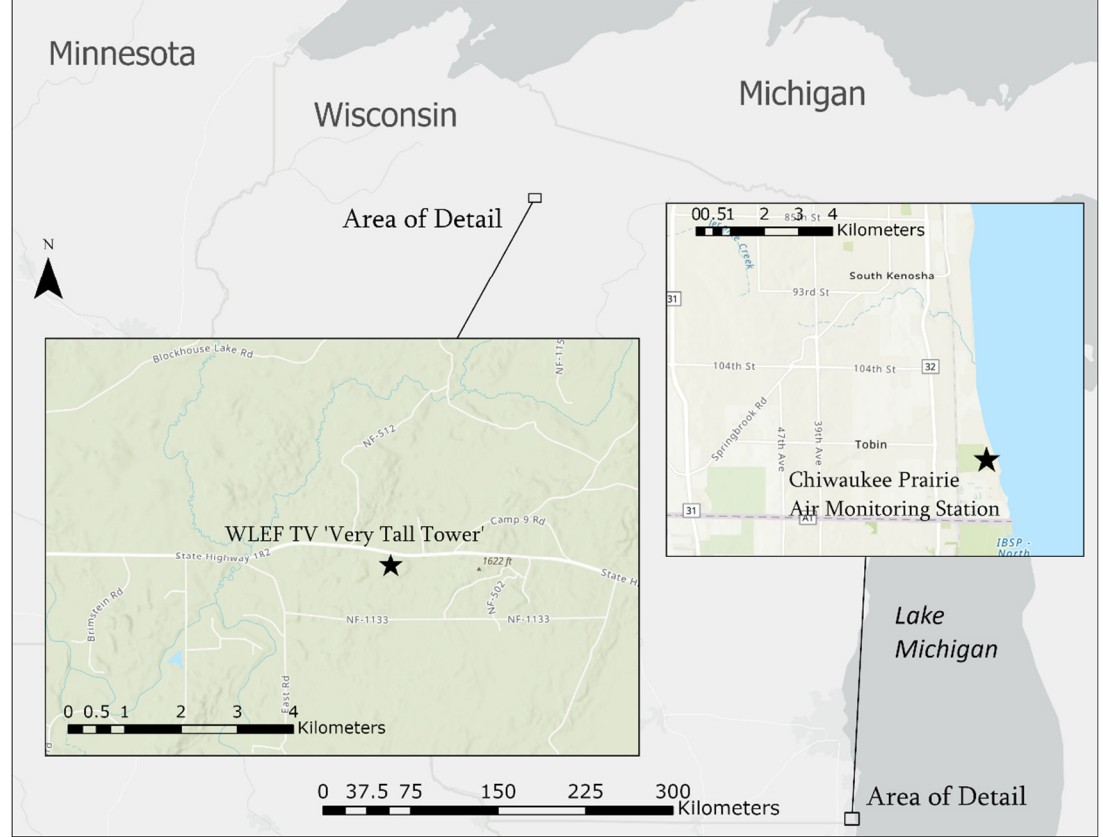

**Figure 1: During the CHEESEHEAD19 and PEcorINO campaigns, measurements were taken at the WLEF TV 'Very Tall Tower'**
**(northern Wisconsin). During the WISCODisco20 campaign, measurements were taken by the Chiwaukee Prairie Air Monitoring**
**Station (southeastern Wisconsin). Map was made with ArcPro 2.8 using ESRI basemap data.**

A follow-up study Probing Ecosystem Responses Involving Notable Organics (PEcorINO) (Vermeuel et al., 2023) was
conducted in September 2020 at the WLEF tower with observations of VOC and ozone at the 30 m inlet (Figure 1). A high-
resolution proton-transfer reaction time-of-flight mass spectrometer (HR-PTR-ToFMS) (Vocus; Aerodyne Research Inc. and
Tofwerk AG (Krechmer et al., 2018)), collected continuous 10-Hz measurements of VOCs and a photometric analyzer
(ThermoScientific 49i) collected 1 Hz $O_3$ measurements at 30 m. Routine US-PFa site measurements of 10 Hz temperature
and 1-Hz measurements of relative humidity (HMP-45C) were also collected during this period.
For CHEESEHEAD19, the Yuneec Typhoon H hexa-copter UAS was flown four days in July 2019 (July 8, 11, 12, and 16)
during the campaign at the WLEF tower and September 13 and 14, 2020 during PEcoRINO. The Typhoon H, owned by UW-
Eau Claire, was chosen for this campaign because it was an inexpensive commercial UAS with capability of holding the
payload of the POM. Flights in 2019 were conducted in the time window of 11 am - 3 pm local time (CDT) and at 11 am and

6 pm CDT in 2020. The Typhoon H was equipped with the POM for each of the flights at the tall tower, and an Intermet Systems meteorological sensor, the iMet-XQ2, for the flights on July 16, 2019, and September flights from 2020 (See Figure 2a). The iMet-XQ2 sensor was placed on the landing gear of the Typhoon H to balance the payload. The days were chosen for suitable flying conditions without strong winds (< 15 mph gusts) or rainstorms or other precipitation. The Typhoon H was flown from a location roughly 100 ft from the tall tower in different patterns to hover for 5 min at 30 m, 60 m, 90 m and 122 m above ground level. Tower gradient uncertainties were determined from 1 standard deviation of the data from 30 and 122 m. The instruments sampling at the 122 m and 30 m heights from the tall tower were switched periodically (Vermeuel et al., 2021). The POM ozone data were collected at 10 s intervals and averaged to 5 minutes.

Before the CHEESEHEAD19 campaign, numerous test flights were necessary to work out payload distribution and to devise flight strategies. The Typhoon H had an approximate 15-minute flight time per battery with the payload. Each flight of the Typhoon H flights consisted of 2 hovers at different heights for 5 minutes. UAS flight log data were saved and was used as primary source for GPS data. All UAS flights were conducted under Federal Aviation Administration (FAA) Part 107 UAS regulations with a licensed UAS pilot.

**2.2 The WiscoDISCO20 Campaign**

The purpose of the Wisconsin's Dynamic Influence of Shoreline Circulations on Ozone (WiscoDISCO) campaign was to investigate the marine inversion influence on ozone measurements at the Lake Michigan shoreline by using an UAS at Chiwaukee Prairie Natural Area in Kenosha County, WI. A regulatory monitor at Chiwaukee Prairie managed by the Wisconsin Department of Natural Resources (WiDNR) records some of the highest ozone in the state of Wisconsin and many Wisconsin shoreline Lake Michigan counties are in nonattainment of federal ozone standards (Stanier et al., 2021). Chiwaukee Prairie is located at the border between Wisconsin and Illinois and is situated between the coastal communities of Winthrop Harbor, Illinois, and Pleasant Prairie, Wisconsin. Suburban housing developments and mixed farmland surround the prairie (Figure 1). Local automobile traffic near to the monitor and UAS launch site was light and limited to neighborhood traffic and occasional train traffic.

The main goal of this campaign was to capture ozone exceedance days at this site where there was an influence of the lake breeze circulation. Ozone exceedance days are typically those in which the synoptic winds bring air from the south northward with high pressure systems over the Ohio Valley (Hanna and Chang, 1995), which are influenced heavily by Chicago pollution plumes. In this environment, temperature inversions commonly form when near-surface air is chilled by thermal exchange with the comparatively cold water of Lake Michigan and are exacerbated when lake breezes advect this dense but shallow layer of cold air inland (Wagner et al., 2022). The result is a shallow pool of colder, denser air overlain by warmer air aloft, with the inversion defined by the temperature increase with height at the boundary between the dissimilar air masses. Inversions act as a cap on the vertical mixing of air that would otherwise dilute and disperse $NO_x$ and VOCs within these pollution plumes. Thus, these ozone precursors can accumulate in the near-surface air to relatively high concentrations.

178       During WiscoDISCO20 UAS were deployed on June 8, 9 and 15-19, 2020. The WiscoDISCO20 campaign was in

collaboration with the Wisconsin Department of Natural Resources' (DNR)'s enhanced monitoring plan for the Chiwaukee
Prairie site and included Pandora (Herman et al., 2009) (a ground-based differential optical absorption spectrometer which
uses the sun as a light source to obtain total column trace gas measurements) and Doppler lidar observations at the site, provided
by the Space Science and Engineering Center at the University of Wisconsin-Madison. The Doppler lidar instruments were
deployed on June 9, 2020 and operated continuously throughout the summer. The Pandora instrument is part of the Pandonia
Global Network, (Verhoelst et al., 2021) which provides automated measurements of total column and tropospheric column
$NO_2$.
A DJI M600 hexa-copter was utilized in a collaborative research endeavor with Purdue University for the WiscoDISCO20
campaign with an FAA compliant Part 107 UAS pilot, Joe Hupy. The DJI M600 had an increased payload capacity with its
camera removed and the ability to place a top-mount for the sensor package, thus increasing the stability of the payload and
providing a longer flight time than the Typhoon H (See Figure 2b). A 3D printed bracket to support the POM was mounted
to the top of the vehicle. The inlet filter cartridge for the POM was held at a position with the least influence from propeller
wash at the center of the top position of the UAS with a ~6 cm inlet tube. The iMet-XQ2 sensor was mounted to the bracket
and secured with cable ties (See SI: Figure S2). During WiscoDISCO20, a series of flights were conducted to produce an
atmospheric vertical profile with fixed altitudes where the UAS hovered for 5 minutes at each designated altitude. The flight
times were approximately 15-20 minutes where the UAS would ascend for 15 m altitude increments where it would hover
for 5 minutes. In an approximate 1.25-hour time window, 8 heights were sampled from 0-122 m AGL with 3 individual
flights (See SI: Table S1). Flights were conducted from a gravel road inside of the Chiwaukee Prairie State Natural Area,
with two focused vertical profile sampling periods: one in the morning at approximately 7-9 am local time (CDT) and one in
the afternoon at approximately 2-4 pm local time (CDT).

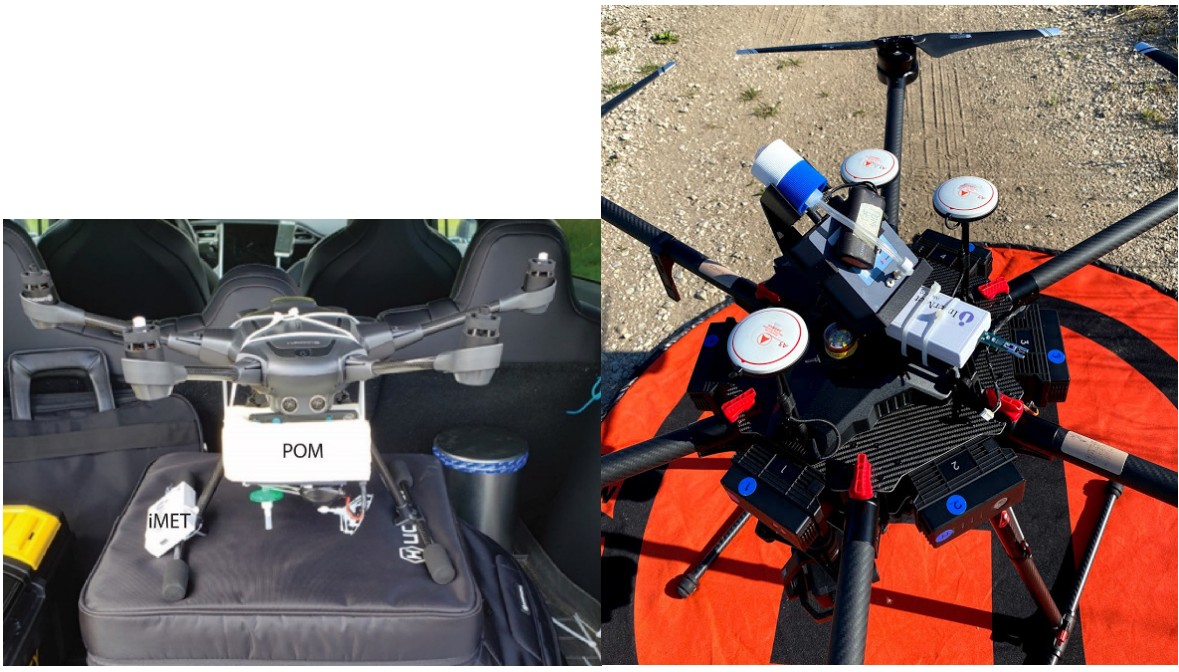


**Figure 2**. a) Typhoon H UAS with mounted POM and iMet-XQ2 as flown during CHEESEHEAD-19 in 2019. POM was housed in foam for vibration dampening. b) Top mounted iMet-XQ2 and POM on a DJI M600 as flown in Sept. 2020 for WiscoDISCO20. The inlet to the POM is held up with a bracket to hold the inlet filter assembly (blue and white).


### 2.3 Personal Ozone Monitor, POM


The 2B Tech personal ozone monitor, POM, measured ozone concentrations via UV absorption spectroscopy. which is
designed to account for a known interference with humidity in the atmosphere (Wilson and Birks, 2006).. The POM measures
ozone concentrations by calculating the difference in absorption between a whole air and a scrubbed ozone air sample in-series
with one optical cell. The POM operates with an in-series duty cycle of measuring the whole air sample for 5 seconds and an
ozone-scrubbed background air sample for another 5 seconds in the same optical cell (Andersen et al., 2010). This differs from
current robust ground analysers, such as the ThermoScientific 49i, which use dual optical cells, one chamber of whole air and
another chamber with ozone scrubbed out to measure a real-time background interference in the absorption signal(Ollison et
al., 2013) (Wilson and Birks, 2006)This duty cycle must be considered when the POM is on a moving platform, as the air
sampled in the first 5 seconds must be representative of the air sampled in the second 5 seconds for each measurement, therefore
slow movement of the UAS during sampling was preferred. For all measurements described here, the UAS was held at one
altitude for 5 minutes to collect representative data from that airmass. The absorption spectroscopy principle behind the POM

with the active background humidity subtraction has a higher specificity to ozone than other light-weight electrochemical sensors(Kim et al., 2018).  The POM was calibrated with the 2B Tech Model 309 transfer standard ozone generator within 24 hours of UAS flights during CHEESEHEAD19. The POM was placed in a foam case to dampen any vibrations associated with the UAS to which it was fastened. The filter on the POM was used for all flights to protect the optical cell from atmospheric particles and debris. The POM was independently powered by lithium-ion batteries. During WiscoDISCO20, the POM was calibrated with the Model 309 ozone generator within 2 hours of each atmospheric profile series of UAS flights. Zero drift of the POM was monitored by collecting scrubbed-ozone data using a cartridge ozone scrubber in between flights. The 2B Tech listed POM accuracy and precision are given as 1.5 ppb or 2% of observations whichever is highest. For the range of observations in this study, the accuracy and precision ranged from 1.5 ppb for many morning observations to up to 2.1 ppb for high ozone afternoon observations.

**2.4 iMet-XQ2**

The iMet-XQ2 sensor is lightweight and portable which allows it to measure temperature (bead thermistor), relative humidity (capacitive), and pressure (piezoresistive) along with recording GPS data with its own internal storage and power systems. The International Met Systems listed iMet-XQ2 accuracy and resolution of ±0.3 °C and 0.01 °C for temperature, ±5% and 0.1% for relative humidity, ± 1.5 hPa and 0.01 hPa for pressure, and an accuracy of 12 m for vertical GPS data. The data were extracted from the iMet-XQS after each flight.

Previous studies have evaluated the accuracy of the iMet-XQ2 on UAS platforms (Kimball et al., 2020; Inoue and Sato, 2022). Kimball et al. executed an exhaustive study on the performance of the iMet-XQ on a UAS in certain solar radiation and wind speed conditions. While they found that under low solar radiation, the accuracy and precision of the temperature measurement followed the listed accuracy and precision, with some direct solar radiation, higher wind speeds on the thermistor allowed for improved precision of the measurements. In cold conditions, shielding the thermistor from both solar radiation and heating from the UAS is important (Inoue and Sato, 2022). Sensor position on the UAS has been found to be important for preventing additional bias from motor heating and propellor wash if the sensor is placed too close to UAS motors (Greene et al., 2019). For this study, a lower accuracy of the iMet-XQ was considered reasonable to ascertain the vertical profile structure of the atmosphere at a shoreline location, if the solar radiation conditions and flight conditions were similar throughout the data collection flight.

**3 Results and Discussion**

**3.1 UAS to Tower Comparisons**

During the CHEESEHEAD19 campaign, an intercomparison was conducted between the observations of ozone from the WLEF tall tower and UAS.. The tall tower ozone measurements were from either a ThermoFischer 49i photometric analyzer or a CI-ToFMS instrument. Each sampled air from either the 122 m or 30 m inlet simultaneously, and source inlets (i.e. sampled

heights) were switched between instruments periodically. The absolute ozone concentrations at the 122 m and 30 m heights from the tall tower ranged from mid-day highs of 40-60 ppb. Tower ozone gradients were calculated as the difference between the ozone measured at 122 m and 30 m inlet heights. These tower observations were compared to the gradient ozone observations made by hovering the UAS at the 122 m and 30 m altitudes for 5 minutes each. The gradient ozone observations reproduced the reported ozone gradients on the tall tower within the considerable uncertainty (See Table 1). The absolute concentrations from the POM were as much as 20 ppb higher than the tower observations (See SI: Figure S3), with tower observations from both the 49i and TOF considered to be the more reliable measurements with consistent calibration procedures and low detection limits (Vermeuel et al., 2021; Vermeuel et al., 2023). Technically the overall comparison between tower gradients and UAS gradients show agreement; however, the considerable uncertainties make POM gradients from 8 and 11 July indistinguishable from zero (See Table 1). This evaluation demonstrated a likely source of inaccuracy with POM ozone observations, with significant offset from the absolute tower observations. These inaccuracies have since been attributed to zero-point drift of the POM, which was substantiated by further laboratory experiments on calibration conditions of the POM. Those experiments showed differences in calibrations due to different sources of power to the POM (batteries versus wall-power). Large noise in the POM observations was attributed to disrupted airflow from propeller wash which was exacerbated by the bottom-mount of the POM on the UAS, as observed as higher noise during take-off and at the start of every hover.

Improvements to the UAS sensor package for the WiscoDISCO20 system were developed as a result of these findings as follows: a) the POM was mounted at the top of a larger UAS with the inlet positioned to the center of a larger, more robust platform, b) the POM was calibrated with the same independent POM battery source as the flights and calibrations were conducted within 2 hours of every flight and c) zero drift was monitored by placing an in-line ozone scrubber on the POM inlet directly after each flight for 5 minutes. The temperature and relative humidity measurements observed from the UAS using the iMet were found to vary from the tower measurements by no more than 1.7°C for temperature and 8% RH (See Table 2).

**Table 1: Comparison of ozone gradients made from Tall Tower at Park Falls and UAS-based POM during CHEESEHEAD-19. Ozone gradient, $\Delta O_3$, calculated as measured $O_3$ at 122 m – $O_3$ at 30 m. The tower measurements were selected as coincident with UAS-mounted POM measurements. Uncertainties reported are propagated from 1 standard deviation at each altitude.**

| Day-Month-Year of Flight | POM UAS $\Delta O_3 \pm \sigma$ (ppb) | Tower $\Delta O_3 \pm \sigma$ (ppb) |
|---|---|---|
| 08-Jul-19 | -5.9 ± 9.6 | 1.0 ± 1.1 |
| 11-Jul-19 | 11.9 ± 21.7 | 8.7 ± 0.8 |
| 12-Jul-19 | 16.1 ± 13.2 | 9.1±1.3 |

**Table 2: Comparison of average air temperature and relative humidity made from Tall Tower at Park Falls and iMet-XQ during CHEESEHEAD-19 and PEcorINO in 2020. The average Tower temperatures at the 30-meter inlet were computed at the time intervals when the UAS altitude was 30 meters AGL. The iMET and Tower data were evaluated at 1 Hz, therefore approximately n=300 for each 5-min hover period.**

| Day-Month-Year of Flight | Altitude A (meters) | iMet UAS T ± σ (°C) | Tower T ± σ (°C) | iMet UAS RH ± σ (%) | Tower RH ± σ (%) |
|---|---|---|---|---|---|
| 16-Jul-19 | 30 | 25.0 ± 0.4 | 24.43 ± 0.07 | 61.2 ± 1.3 | 66.8 ± 0.4 |
| 13-Sep-20 | 30 | 15.5 ± 0.3 | 13.8 ± 0.9 | 68.9 ± 0.8 | 76.7 ± 5.5 |
| 14-Sep-20 | 30 | 13.5 ± 0.8 | 13.0 ± 0.8 | 63.0 ± 6.4 | 61.5 ± 6.8 |

### 3.2 Observations at Lake Michigan Shoreline: WiscoDISCO20 UAS to Ground Comparisons

The viability for UAS-mounted ozone observations to capture low-altitude features in ozone is well-matched to the small-scale vertical structure of marine layer ozone concentrations in a near-shore environment. For the June 2020 observations, the UAS platform was the DJI M600 with a top-mounted bracket for positioning the filter cartridge for the POM in a space for minimal disruption of the air mass from propeller wash. The iMet-XQ2 sensor was mounted to this bracket to the side of the POM (See SI: Figure S2). The DJI M600 was flown at the Chiwaukee Prairie State Natural Area to capture shoreline airmasses impacted by lake breeze onshore flow during time of high ozone. The week of June 15-19, 2020, provided ideal conditions for high ozone and lake breeze conditions (as seen in Figure 3) where daytime winds shifted regularly to southeasterly and daily maximum temperatures increased throughout the week (See SI for identification of lake breeze from GOES-East satellite imagery). Most days during the week of June 15-19 had observable cumulous cloud suppression fronts over land near to the shoreline of Lake Michigan which is indicative of marine air incursion over land (see SI: Figures S4-S5). Particulate matter concentrations also increased during the week. The UAS was flown in a 2-hour window to capture morning and afternoon

vertical atmospheric profiles. A single battery flight of the UAS accounted for 3-4 hover heights and multiple sets of batteries
were used to hover for 10 total points to get a vertical distribution.

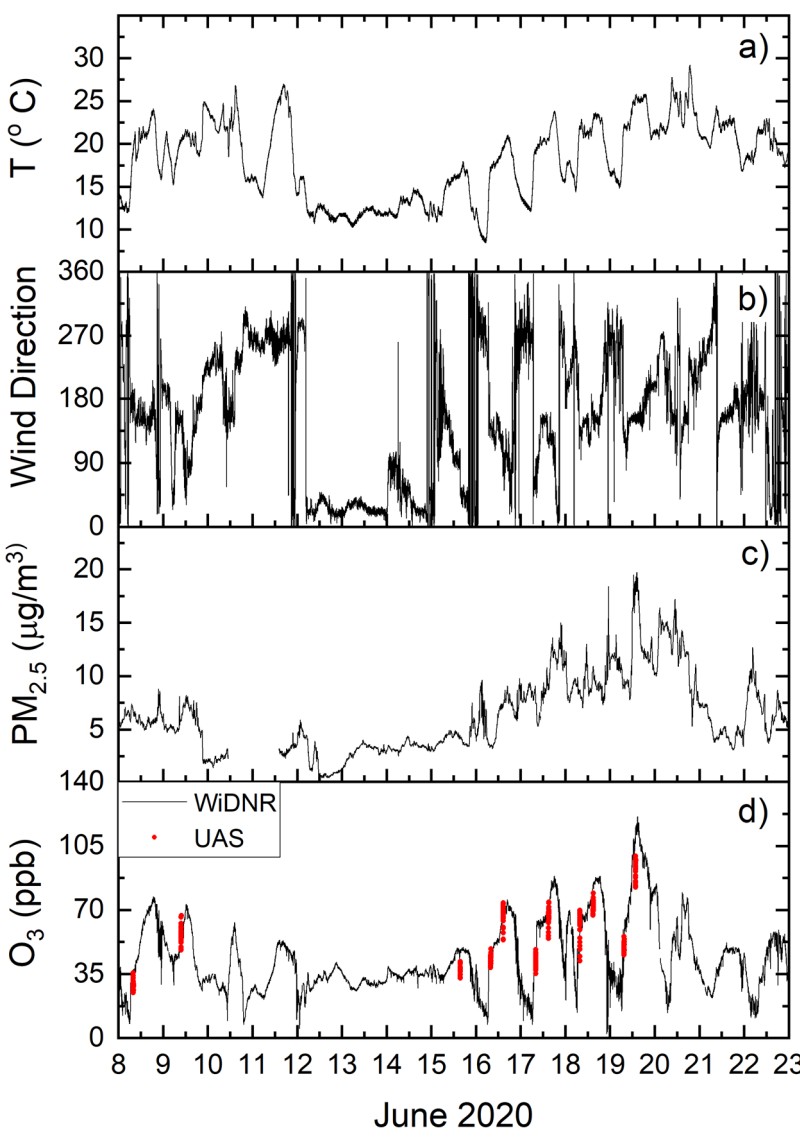


**Figure 3: June 8-22, 2020 a) temperature in ºC b) wind direction, c) PM$_{2.5}$ in µg/m$^3$ and d) O$_3$ as measured at the WDNR ground station (black) and on the UAS via POM (red).**

The accuracy of the ozone concentration, temperature, and relative humidity (RH) observations made aloft on the UAS was
evaluated by comparing the lowest altitude hover altitude at 9 meters above ground level (m AGL) to 1-minute data from the
local air monitoring station in Chiwaukee Prairie (AIRSID# 55-059-0019). The uncertainty in the UAS-mounted POM was
determined to be the 1 standard deviation in the averaged 10 s data. A regression analysis of the two observations is given in
Figure 4a; these data are strongly correlated as the $R^2$ value is 0.939. The linear fit to the graph is weighted by the highest
ozone data and the RMSD = 5.3 ppb. Some disagreement could be from the discrepancy in altitudes for the two observations
(the DNR inlet is at 4.5 m in comparison to the first altitude for hovers at 9 m), or to accuracy issues with the zero drift as
observed during CHEESEHEAD-19. A similar comparison was conducted for the iMet temperature measured at the lowest
hovering altitude (approx. 9 m) with ground temperatures (Figure 4b) with an agreement at $R^2= 0.944$. With these added
observations, the accuracy for the $O_3$ concentrations via UAS-mounted POM are considered accurate within 10 ppb, and the
iMet temperatures within 11 %. This figure has some similarities for the Li et al. (Li et al., 2020) figure 5a, where they saw a
linear fit of 0.7x – 7 for a POM correlation to a regulatory ozone measurement instrument standard. The difference between
our measurement and theirs is that we see more observations along the 1:1 line with higher ozone concentrations deviating the
most from the center line, whereas Li et al. (2020) showed a consistent linear response at ~70% of the regulatory $O_3$
measurement.

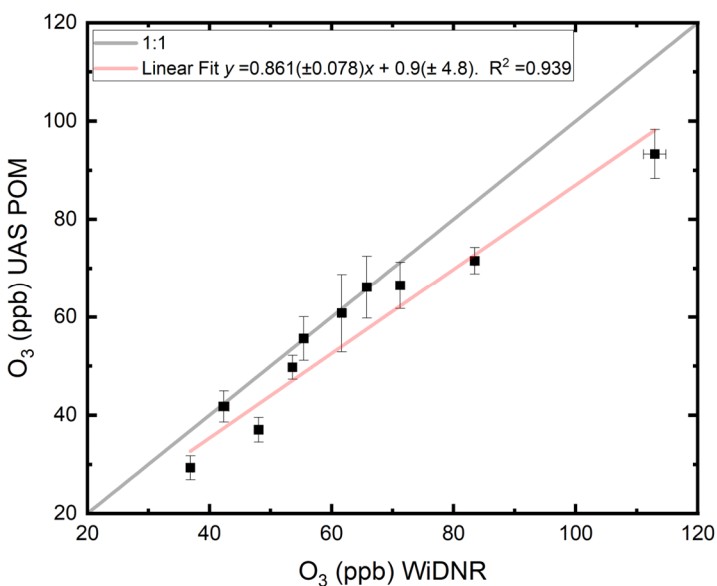

311                    a)

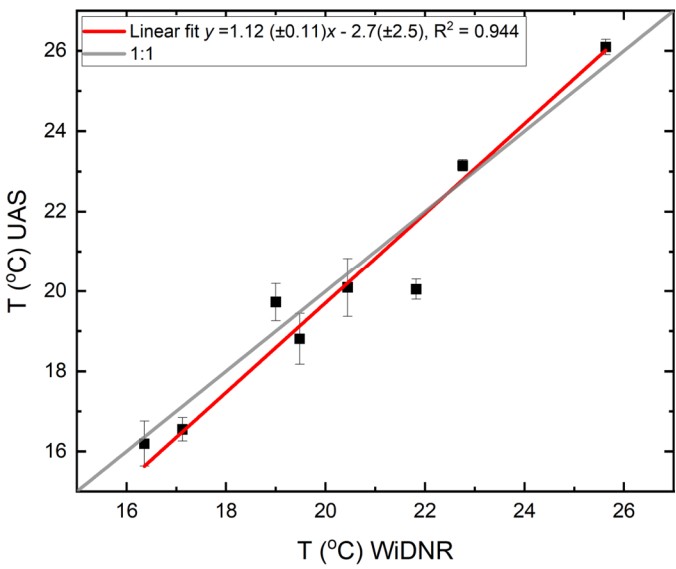

312        b)


**Figure 4: a) Intercomparison O₃ UAS POM measurements in comparison to O₃ WiDNR Chiwaukee Prairie measurements on June 8, 9, 15-19 2020. The 5- minute average WiDNR and UAS POM data from the lowest hovering altitude is with uncertainties as 1σ from the mean. The grey line demonstrates a 1:1 line and the red line depicts a linear regression fit (R² = 0.939), with a fit of [O₃ POM ]= 0.861 (±0.078) [O₃ WiDNR ] + 0.9 (± 4.8). b) Intercomparison of temperature from lowest altitude reading from the UAS-mounted iMet-XQ2 and the WiDNR ground station. Red line indicates the linear regression (T iMET-XQ2  = 1.12 (±0.11) T WiDNR - 2.7(±2.5), R² = 0.944) and the gray line is 1:1 fit.**

320

### 3.3 Case study: Low Altitude Gradients at the Lake Michigan Shoreline

The week of June 15-19, 2020 had 4 days where O₃ concentrations exceeded 70 ppb (Figure 3 d). The dominant winds were from the south and lake breezes were observed on all days that week (Figure 3 b), which are conditions conducive to higher ozone concentrations along the Lake Michigan shoreline due to Chicago emissions getting trapped over Lake Michigan during optimal conditions for photochemical production of ozone and then advecting ozone back on land at the shoreline (Vermeuel et al., 2019; Abdi-Oskouei et al., 2020; Cleary et al., 2022b; Baker et al., 2023). The conditions near Lake Michigan were consistently sunny at the shoreline with some evidence for cumulus cloud formation inland on June 19, 2020 often used as a identifying signature of lake breeze from satellite observations (Levy et al., 2010; Sills et al., 2011).

Vertical profiles for UAS flights were constructed using hovering altitudes of the UAS and time stamps for each altitude to determine observed average O₃, temperature, pressure and relative humidity (RH) for each altitude. Because of limited battery time, each vertical profile was constructed from 3-4 hovering altitudes during 3 separate 20-minute flights, covering a time window of approximately 1.25 hours (See SI Table S2). Figure 5 depicts vertical profiles of potential temperature overlaid

with ozone concentrations through the week of June 15-19, 2020. Every day shows an inverted stable atmosphere. Some days show a well-mixed buoyant internal boundary layer in the lowest 40-100 m AGL (Figure 5) where the potential temperature is close to a vertical line with respect to altitude, particularly in the June 18 and 19 afternoon flights. This discontinuity of most vertical profiles exists where the lowest 40-60 m AGL is closer to a more vertical potential temperature profile, which we refer to as the internal boundary layer, followed by a steep inversion aloft, most pronounced in June 16, 17, and 18 afternoons with a gradient of 5 K or more within 60-100 m AGL. The morning of June 18 (Fig 5-c) was the only day to show a steep inversion down to the surface with no discontinuity. Ozone concentrations in all ascents had maximum observations below the maximum altitude of the flight. Ozone concentrations tended to be highest near areas of steep inversion (June 15, June 17 am and pm, and June 18 pm flights) or near/within the internal boundary layer (June 16 pm, June 19) except on June 18 in the morning when ozone concentrations were highest at the lowest altitudes when the inversion extended to the surface. For all 5 days, observed afternoon maximum ozone concentrations were observed at higher altitudes than adjacent to the surface (Figures 5 a-e). The higher ozone concentrations in the vertical profiles tended to be at or near the maximum inversion generally in the region of 40-60 m AGL.

The variation in height of the steep inversion layer is evident in the day-to-day differences, from as low as 40 m AGL (June 15, 16, and 17) to as high as 100 m AGL on June 19. Morning to afternoon differences on July 16 and 17 show a steeper gradient in temperature later in the afternoon with not much change in the inversion height. By contrast, on the morning of July 18, the vertical profile of temperature shows an inversion starting at the surface (Figure 5d) and by the afternoon the inversion height starts at 60 m AGL. In comparison, turbulent kinetic energy (TKE) based boundary layer depths given by the High-Resolution Rapid Refresh (HRRR) (Dowell et al., 2022) atmospheric model outputs extend from 80 to 250 m AGL for this location, not as low as the data in Figure 5. HRRR boundary layer height is a metric which addresses how photochemical models are treating vertical profiles when computing photochemical ozone production. The use of the HRRR boundary layer height highlights the sub-grid scale of the vertical profiling, which indicates that UAS observations can sample important properties of the marine air incursion of a lake breeze. The lower boundary layer heights in the afternoon in comparison to the morning are proposed to arise from stronger synoptic winds degrading the inversion from above (Lyman and Tran, 2015). Doppler lidar measurements (which cannot make observations below 100 m AGL) show high aerosol loading in the afternoons at altitudes close to the ~100 m cut-off altitudes below which the instrument has a dead zone, which may correspond to continued inversion up to 200 m or more. The UAS observations give a complementary measurement to indicate the region of inversion and the compositional layering below, within, and above the inversion to demonstrate a more complicated picture of mixing and vertical stratification in the lower atmosphere. While these measurements may not adequately address exactly why models do not represent the shoreline effectively (See SI Figure S6), they can shed light on the required resolution and vertical structure that encompasses plume volume within an inverted atmosphere near Lake Michigan.

The UAS observations at Chiwaukee Prairie shown here demonstrate a very shallow internal boundary layer (40-100 m AGL) developed over land underneath the temperature inversion (modeled boundary layer heights 80-250 m AGL), where ozone is found to be in highest abundance near the maximum inversion. On two days with the highest internal boundary layer

height, ozone concentrations were highest within the internal boundary layer, suggesting that an extended internal boundary layer height over land could possibly play a role in transport of pollutants in the marine layer. However, more observations of atmospheric profiles of ozone and meteorology are required over land and over water to better characterize this transitional environment.

The work by Li et al. (2020) described use of POM and particle observation on a fixed-wing UAS flying at a speed of 150 km/hr and compared measurements from those instruments to regulatory instruments on a tethered airship and addressed intercomparison with the POM and a regulatory ozone measurement instrument ($O_3$42M from ESA). They used an insulated box for the POM and were able to show high correlation with a regulatory monitor, but with an offset. Their conclusions are that the POM measures atmospheric variability consistent with a regulatory monitor but demonstrates a negative bias. Here, we flew the POM at a much lower flight speed, and only averaged data from a single hovered point at which we stayed for 5 minutes each flight. This was to address the duty-cycle limitations of the POM with the on-off in series subtraction of the water vapor absorption. Li et al. address only that the regulatory monitor they used for comparison which employed an in-line heating method for removing water vapor interference, instead of a dual-cell active subtraction in parallel as is typical for other regulatory monitors. While Li et al. 2020 demonstrated some correlation between RH and variability between the UAS-mounted POM and tethered-airship-platform regulatory monitors, they do show that vertical gradients can be captured by UAS and tethered airship, but with discrepancies in location of planetary boundary layer. This is consistent with our observations that the gradient observations from UAS are in agreement (with high variability) with tower-based observations in the lowest 120 m AGL. What we cannot account for here is the difference in POM variability on a UAS which hovers for 5 minutes in comparison to a fixed-wing travelling at 150 km hr$^{-1}$, which may also lead to additional variability in the measurement due to inlet pressure changes and optical cell vibrations. Additional improvements to the POM performance could arise from a) thermal insulation and b) a slow constant accent instead of hovering and are intended for future studies. Additional improvements to the iMET-XQ2 performance could arise from a slow ascent (to assist in aspirating the thermistor) and shielding the iMET to account for solar irradiation of the sensors.

The feasibility of using UAS in shoreline environments depends on the vertical scale of the phenomenon, the UAS flight time and requisite battery life for such UAS observations and the legal flight conditions within each municipality. Some researchers have successfully used UAS for vertical ozone profiles up to 1000 m (Chen et al., 2022; Wu et al., 2021), tethered balloons (Li et al., 2020; Chen et al., 2022) and thermally insulated a UAS-mounted POM in the winter (Chen et al., 2020; Chen et al., 2019). The scales of sea breeze influence on vertical profiles have ranged from 400-600 m AGL at coastal locations in Nova Scotia (Gong et al., 2000), 600-800 m AGL at coastal locations in China (Wu et al., 2010) and 400-800 m AGL in lake breeze influenced locations in Saskatchewan (Sun et al., 1998). The lake breeze vertical dimensionality near Lake Michigan has been shown to have large effects at altitudes from 50-500 m AGL from crewed aircraft (Stanier et al., 2021; Cleary et al., 2022b), remote sensing (Wagner et al., 2022) and UAS studies (Tirado et al., 2023).

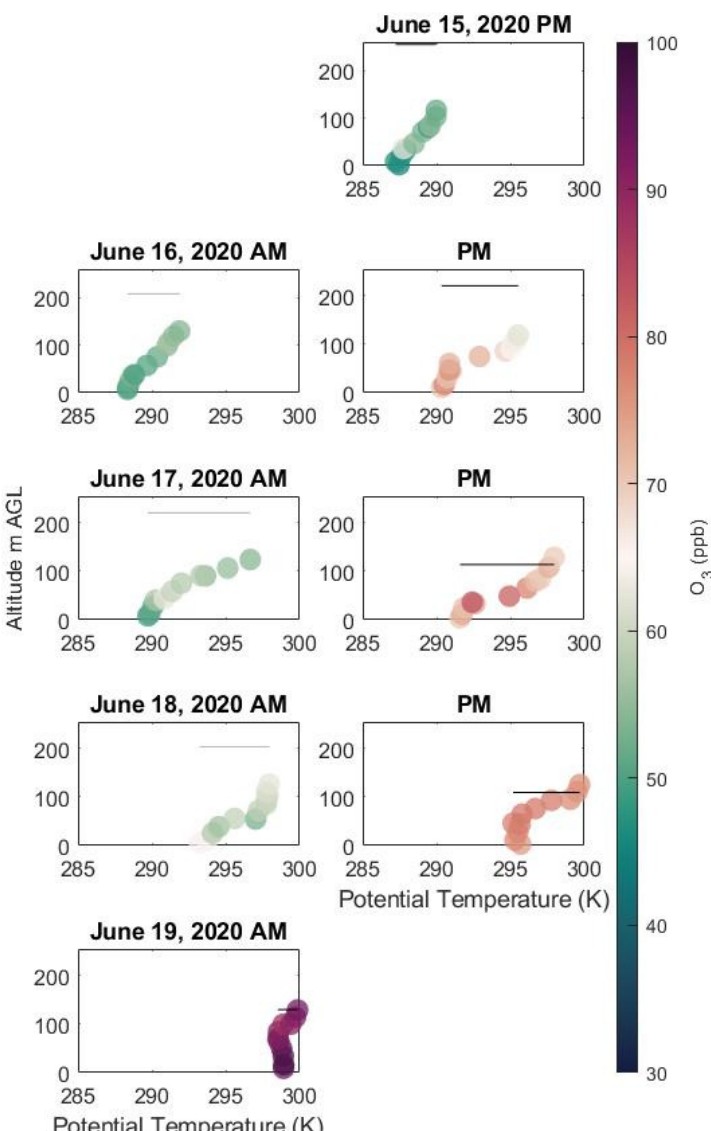

400

**Figure 5: Altitude (m AGL) versus potential temperature (K) with O₃ (ppb) for a) June 15 flights, June 16 b) morning, and c) afternoon flights, June 17 d) morning and e) afternoon flights, June 18 f) morning and g) afternoon flights, and h) June 19 afternoon flights. All times in CDT (2020). Grey and black bars indicate HRRR boundary layer heights for morning and afternoon, respectively.**

405

## 4 Conclusions

An UAS atmospheric profiler was tested in comparison to tower-based instrumental observations indicating a need for careful adjustments to operating parameters for the ozone measurements. Improvements including a top-mount for the sensor package, a larger UAS and higher frequency calibrations and zero-drift checks were shown to improve overall accuracy of the ozone observations near a ground air monitoring station. The improved vertical atmospheric profiler was shown to capture atmospheric variability in the lowest 120 m of the atmosphere at a Lake Michigan shoreline region, demonstrating a feasible use for UAS in atmospheric sampling to connect the scales of ground-based observations and tower or remote sensing aloft. These observations are the first UAS observations of ozone near Lake Michigan that document the over-land penetration of the marine layer and gradients in ozone within it. This work highlights the necessity for higher vertical resolution in observations in this shoreline location to inform improvements to how air quality models represent the stratification and mixing of air parcels at locations like these.

Suggestions for further improvements: in this study, the POM performance on UAS was improved by inlet positioning and slow flight parameters, top-mount placement on a robust UAS and increasing the rate of calibrations to pair each calibration with specific battery power source improved the precision and accuracy. However, added thermal insulation, as described by Li et al, appears another promising additional consideration for improved performance of the POM on UAS. The POM appears to be a robust enough instrument for course atmospheric measurements in the atmosphere (to 2 ppb precision or 2% of reading) but integration onto a UAS should be carefully considered. A wider variety of studies have been conducted on iMET performance on UAS (Kimball et al., 2020; Inoue and Sato, 2022; Greene et al., 2019).

## Appendices

Supplemental Info

## Data Availability

A data repository was generated for CHEESEHEAD19 at: https://data.eol.ucar.edu/master_lists/generated/cheesehead/ (last accessed 7/6/2023).

A data repository was generated for the WiscoDISCO 2020 field campaign at: https://zenodo.org/communities/wiscodisco2020/ (last accessed 7/6/2023). Dataset available at DOI: 10.5281/zenodo.8118176. Each data file is in a .txt tab delimited structure with descriptive column titles. Any data file with a full suite of data from both iMET and POM instruments is given without a qualifier. On days when data was collected from one of the instruments, the file names indicate them as such.

## Author Contributions

JR, BK, MPV and KK contributed to data acquisition, data analysis and manuscript writing. WM, SZ, and AV contributed to data analysis. GP and TW contributed to data acquisition and manuscript writing and editing. AD, TB, and RBP contributed to field campaign planning and manuscript editing and writing. JPH contributed to field campaign planning, data analysis and manuscript writing and editing. PAC contributed to field campaign management, data acquisition, data analysis, manuscript writing and editing.

## Competing Interests

The contact author has declared that none of the authors has any competing interests.

## Acknowledgements

University of Wisconsin -Eau Claire team acknowledges funding from the Office of Research and Sponsored Programs Faculty/Student Research Collaboration Grants through Blugold differential tuition. This research is funded through the National Science Foundation Grant AGS-1918850. Ankur Desai coordinated the efforts for CHEESEHEAD19 and acknowledges funding from NSF AGS-1822420 and the Dept of Energy Ameriflux Network Management Program award to the ChEAS core site cluster. Any opinions, findings, and conclusions or recommendations expressed in this material are those of the author(s) and do not necessarily reflect the views of the National Science Foundation.  Ancestral & Indigenous Lands Acknowledgement: We acknowledge that our research took place at the ancestral lands of the Očhéthi Šakówiŋ, Anishinabewaki, Ho-Chunk, Myaamia, Potawatomi, Kaskaskia, Peoria and Kiikaapoi people.

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
