# Peer review of "Observing Low Altitude Features in Ozone Concentrations in a Shoreline Environment via Uncrewed Aerial Systems"

_Atmospheric Measurement Techniques, 2023_

## Referee Comment (RC1)

**Reviewer comments for "Observing Low Altitude Features in Ozone Concentrations in a Shoreline Environment via Unmanned Aerial Systems"**

Josie K. Radtke et al.
*Atmos. Meas. Tech.*

amt-2023-143

**Recommendation: Reject**

**General Comments**

This paper presents a summary of uncrewed aircraft system (UAS) flights performed during the CHEESEHEAD19, PEcorINO, and WiscoDISCO20 campaigns that collected vertical profiles of atmospheric ozone measurements. The need for these types of measurements in general is well motivated, but the objectives of the paper are not well stated and in general the paper lacks discussion on the unique aspects of the sensor integration, sophisticated comparisons with reference measurements, and outlooks on future applications. While reasonably appropriate in scope for the journal *Atmospheric Measurement Techniques*, in my opinion there are issues with this study's presentation, experimental procedure, and scientific significance that would require substantial revisions before publication.

**Fatal Flaw**

In my opinion, this study is limited in regard to its overall contribution to the scientific literature, particularly in the realm of atmospheric observations with UAS. While the authors did a commendable job discussing the need for low-level observations of ozone in coastal environments, the primary results (a collection of vertical profiles to 120 m altitude with some comparisons to tower-based sensors) of the study are mostly proof-of-concept measurements collected with commercially available UAS airframes and sensors. Typically, these types of studies in AMT are focused on the design of custom-built UAS or unique sensor package integration (e.g., Altstädter et al., 2015; Segales et al., 2020; Hamilton et al., 2022), but that does not seem to be the focus of the present article. Otherwise, to my knowledge, there have already been a handful of studies collecting observations of atmospheric trace gases (including ozone) with UAS in a more systematic manner than the limited selection of cases presented here (e.g., Schuyler and Guzman, 2017; Schuyler et al., 2019; Krautwurst et al.,

2021; Bretschneider et al., 2022, and references therein). Considering these factors, in my opinion the revisions necessary to improve this paper's contributions to the existing literature are too substantial at this time such that the submission should be rejected. However, I do believe the content of this study may warrant submission to a data journal such as *Earth System Science Data* to complement the data repositories cited at the end of the article.

**Major Comments**

1. While the motivation for the study is reasonably established, it is not immediately clear in the introduction what the objectives of this study are outside of generally assessing some UAS vertical profile measurements of ozone. At the end of the introduction, please explicitly outline the relevant scientific questions, hypotheses, and/or novel concepts this paper will present.

2. Although not the primary focus of the study, I am rather concerned with the quality of meteorological observations collected by the iMet sensors due to their siting onboard the UAS airframes. In particular, there have been numerous investigations on the placement of temperature sensors to mitigate the influences of solar radiation, heat from the UAS motors, and heat from the body of the UAS itself while still maintaining adequate ventilation (see the discussions in Greene et al., 2018, 2019; Barbieri et al., 2019; Islam et al., 2019; Kimball et al., 2020). While the iMet-XQ2 mounting onboard the Typhoon H (Figure S1) seems reasonably well sited, the position on the MJ600 (Figure S2) likely resulted in biases due to lack of ventilation and exposure to direct sunlight and heat from the black aircraft body. While this is not something that can be corrected for necessarily, please at least include a discussion on this in the results.

3. In Sections 3.1 and 3.2, UAS measurements are compared with tower-based and ground-based references. However, this is mostly presented as single cases summarized in tables 1 and 2 as well as figure 3. For a calibration procedure, I think a more thorough analysis is warranted, especially to contextualize the cases presented in section 3.3. For example, how many individual data points were collected on each day and at each level? Additionally, bulk statistics across all days such as the mean, median, and standard deviation of the differences between UAS and tower measurements of each variable would be pertinent. What conditions are present where the largest biases are observed?

4. The entire paper is building up towards the results from the cases in Section 3.3, with a lot of emphasis on the vertical distribution of ozone and temperature versus height across multiple days. These results are provided in Figure 4, with the discussion focusing on features of these vertical profiles. The current presentation and layout of this figure, however, make it difficult to follow the discussion in Section 3.3. For example, with emphasis on changes in the vertical, I recommend changing the layout of the subpanels to be organized horizontally instead of being stacked vertically so that the subpanels are taller than they are wide (or at least with an aspect ratio of 1:1). Additionally, it is difficult to tell the difference between the AM and PM profiles for

the June 16, 17, and 18 cases; please consider using different shapes (circles, squares, crosses, etc.) for the different profiles on the same day. I also strongly urge the use of a colorblind friendly color palette that is also uniformly perceptive in place of the current rainbow color bar(see Stauffer et al., 2015). Finally, why was the choice made to use the HRRR PBL height as a reference in this figure? You mention there was a Doppler lidar present for the WiscoDISCO20 campaign, was this capable of producing PBL height estimates more locally? Otherwise, consider omitting the earlier discussions on the instruments not used for this current study.

**Minor and Technical Comments**

1. L26: Please remove the period at the start of the line.

2. L40: Please define the acronym UAS.

3. L42: Please remove the extra period between the citations and the start of the next sentence.

4. L57: Spelling error: should "crate" read as "create"?

5. L86: Please define the acronym "UW" in UW-Eau Claire

6. L93: Spelling error: remove the "F" at the start of the word "and".

7. Section 2.2: Here I have a handful of suggestions for breaking this long first paragraph up into logical sections. First, at L138, the stentence starting "The main goal of this campaign..." could start a new paragraph. Similarly, break a new paragraph as L147 starting with "During WiscoDISCO20 UAS...".

8. 148: Please define the acronym "DNR."

9. L148–154: Did you use these instruments specifically in this study? Consider omitting this portion (see major comment 4).

10. L165: This would be a good place to highlight the total number of flights conducted in each period.

11. L177: Add a space between "electrochemical sensors" and the following parenthetical citation.

12. L194: Should this read "an intercomparison..." instead of just "n"?

13. L202–204: The sentence beginning "The UAS gradient observations..." is a bit hard to follow, please consider rewording.

14. L208: I recommend breaking a new paragraph starting with the sentence "Improvements to the UAS sensor package..."

15. Table 1 and surrounding discussion: This is perhaps semantic, but these are not necessarily gradients but rather just differences. Please consider changing the wording throughout, or computing the gradients by dividing the differences by the height between the sensors.

16. Table 2: Please include the total number of flights and/or individual samples that go into each mean and standard deviation presented here (see also major point 3).

**References**

Altstädter, B., and Coauthors, 2015: ALADINA – an unmanned research aircraft for observing vertical and horizontal distributions of ultrafine particles within the atmospheric boundary layer. *Atmospheric Measurement Techniques*, **8 (4)**, 1627–1639, https://doi.org/10.5194/amt-8-1627-2015.

Barbieri, L., and Coauthors, 2019: Intercomparison of Small Unmanned Aircraft System (sUAS) Measurements for Atmospheric Science during the LAPSE-RATE Campaign. *Sensors*, **19 (9)**, 2179, https://doi.org/10.3390/s19092179.

Bretschneider, L., and Coauthors, 2022: MesSBAR—Multicopter and Instrumentation for Air Quality Research. *Atmosphere*, **13 (4)**, 629, https://doi.org/10.3390/atmos13040629.

Greene, B. R., A. R. Segales, T. M. Bell, E. A. Pillar-Little, and P. B. Chilson, 2019: Environmental and Sensor Integration Influences on Temperature Measurements by Rotary-Wing Unmanned Aircraft Systems. *Sensors*, **19 (6)**, 1470, https://doi.org/10.3390/s19061470.

Greene, B. R., A. R. Segales, S. Waugh, S. Duthoit, and P. B. Chilson, 2018: Considerations for temperature sensor placement on rotary-wing unmanned aircraft systems. *Atmospheric Measurement Techniques*, **11 (10)**, 5519–5530, https://doi.org/10.5194/amt-11-5519-2018.

Hamilton, J., G. de Boer, A. Doddi, and D. A. Lawrence, 2022: The DataHawk2 uncrewed aircraft system for atmospheric research. *Atmospheric Measurement Techniques*, **15 (22)**, 6789–6806, https://doi.org/10.5194/amt-15-6789-2022.

Islam, A., A. L. Houston, A. Shankar, and C. Detweiler, 2019: Design and Evaluation of Sensor Housing for Boundary Layer Profiling Using Multirotors. *Sensors*, **19 (11)**, 2481, https://doi.org/10.3390/s19112481.

Kimball, S. K., C. J. Montalvo, and M. S. Mulekar, 2020: Assessing iMET-XQ Performance and Optimal Placement on a Small Off-the-Shelf, Rotary-Wing UAV, as a Function of Atmospheric Conditions. *Atmosphere*, **11 (6)**, 660, https://doi.org/10.3390/atmos11060660.

Krautwurst, S., and Coauthors, 2021: Quantification of $CH_4$ coal mining emissions in Upper Silesia by passive airborne remote sensing observations with the Methane Airborne MAPper (MAMAP) instrument during the $CO_2$ and Methane (CoMet) campaign. *Atmospheric Chemistry and Physics*, **21 (23)**, 17 345–17 371, https://doi.org/10.5194/acp-21-17345-2021.

Schuyler, T. J., S. C. C. Bailey, and M. I. Guzman, 2019: Monitoring Tropospheric Gases with Small Unmanned Aerial Systems (sUAS) during the Second CLOUDMAP Flight Campaign. *Atmosphere*, **10 (8)**, 434, https://doi.org/10.3390/atmos10080434.

Schuyler, T. J., and M. I. Guzman, 2017: Unmanned Aerial Systems for Monitoring Trace Tropospheric Gases. *Atmosphere*, **8 (10)**, 206, https://doi.org/10.3390/atmos8100206.

Segales, A. R., B. R. Greene, T. M. Bell, W. Doyle, J. J. Martin, E. A. Pillar-Little, and P. B. Chilson, 2020: The CopterSonde: An insight into the development of a smart unmanned aircraft system for atmospheric boundary layer research. *Atmospheric Measurement Techniques*, **13 (5)**, 2833–2848, https://doi.org/10.5194/amt-13-2833-2020.

Stauffer, R., G. J. Mayr, M. Dabernig, and A. Zeileis, 2015: Somewhere Over the Rainbow: How to Make Effective Use of Colors in Meteorological Visualizations. *Bulletin of the American Meteorological Society*, **96 (2)**, 203–216, https://doi.org/10.1175/BAMS-D-13-00155.1.

---

## Author Comment (AC1)

AMT-2023-143 Author's Response to Reviewer 2:

We thank the reviewer for their thorough evaluation of the submitted manuscript and value the suggestions provided to strengthen the paper. Author's comments are in blue.

We can absolutely address the reviewer's concerns regarding the structure of the manuscript to emphasize the techniques described in the manuscript to better highlight the utility of this approach and the improvements to using UAS for atmospheric measurements as was a result of this work and outlined in the document.

Specifically:

In a revised manuscript, we can expand upon the utility and benefits of using a POM and also the difficulties of mounting this instrument on a UAS platform.

We can highlight the literature on iMET on UAS and how this informed the work done here.

We can also address the choice of UAS for these two campaigns. The first UAS, the Typhoon H, was chosen as an inexpensive commercial UAS with capability of holding the payload of the POM. The second UAS, the DJI M120, had an increased payload capacity with its camera removed and the ability to put a top-mount for the sensor package, thus both increasing the stability of the payload and also an increased flight time.

This paper describes an ozone and meteorological measurement system mounted on two different hexacopter UAS, flown over land and near water, and compared with fixed sensors.  UAS measurements in the atmosphere are rapidly becoming more common (as demonstrated in the references cited here), and though they have limitations from instrument weight and power consumption, they could potentially make important measurements of meteorology, atmospheric trace gases and aerosols.  This work can be considered as a step toward progress in this area, especially in terms of the high spatial resolution measurements useful in studies of the boundary layer and lake breeze/land breeze events.  However, the manuscript in its current form is not as informative as it could be, and could use a bit of rewriting.  To make a more meaningful contribution to the literature, I recommend the following changes:

The paper should be restructured to emphasize the measurement technology aspects, both since it is in review at Atmospheric Measurement Techniques, but also because this is really much of the new and useful information contained in the manuscript.  (Although I also agree that section 3.3 has some interesting science in

it.)  First, I would move the paragraph about the Personal Ozone Monitor (POM), now 2.3, to the start of the Materials and Methods and make it 2.1.  The iMet sensor could go right after that.  The authors can decide whether to have a section of its own for the UAS used in the three studies, but right now there is just the bare minimum of description of the two UAS.  Can anything be added to describe why these were chosen, what the necessary characteristics of a UAS for this research are, how they worked as an airframe/sensor package, and how they could be improved?  Also, the introduction could be changed to emphasize more the potential for UAS measurements in the boundary layer or near-shore environment to add to our understanding of chemical composition and atmospheric structure there.  This might only require a few sentences added or rewritten, but it would help the paper become more coherent and targeted.  Finally, in the Results and Discussion section, how do these results compare with the previous experiments of Li et al. 2020 for ozone?

This can be addressed in a revised manuscript by adding the following language:

The work by Li et al. (2020) described use of POM and particle observation on a fixed-wing UAS flying at a speed of 150 km/hr and compared measurements from those instruments to regulatory instruments on a tethered airship and addressed intercomparison with the POM and a regulatory ozone measurement instrument ($O_3$42M from ESA). They used an insulated box for the POM and were able to show high correlation with a regulatory monitor, but with an offset. Their conclusions are that the POM measures atmospheric variability consistent with a regulatory monitor but demonstrates a negative bias. Here, we flew the POM at a much lower flight speed, and only averaged data from a single hovered point at which we stayed for 5 minutes each flight. This was to address the duty-cycle limitations of the POM with the on-off in series subtraction of the water vapor absorption. Li et al address only that the regulatory monitor they used for comparison did a heating method for removing water vapor interference, instead of a dual-cell active subtraction in parallel as is typical for other regulatory monitors. While Li et al 2020 demonstrated some correlation between RH and variability between the UAS-mounted POM and tethered-airship-platform regulatory monitors, they do show that vertical gradients can be captured by UAS and tethered airship, but with discrepancies in location of PBL. This is consistent with our observations that the gradient observations from UAS are consistent (with high variability) with tower-based observations in the lowest 120 m AGL. What we cannot account for here is the difference in POM variability on a UAS which hovers for 5 minutes in comparison to a fixed-wing travelling at 150 km hr$^{-1}$, which may also lead to additional variability in the measurement due to inlet pressure changes and optical cell vibrations.

The POM (because of its very low weight and power consumption) is a very attractive sensor for UAS use, but did it work?  Can it work?  If not, what sensor (from 2B or elsewhere) would be needed, and how much extra weight does that require?  What would need to be changed to optimize the UAS for this kind of experiment?  Again, this should not take a lot of space, but would improve the impact of the manuscript.

We think that Figure 3a from this figure shows a reasonable agreement for POM measurements made aloft and a ground-based observation. Certainly, that agreement is improved from the tower-based comparison. Therefore, the POM on a slow-moving UAS with a high flight-time and inside an insulated box (as described in Li, et al. 2020 and Wang, et al. 2017) is likely the best solution to placing an $O_3$ sensor on a UAS. The electrochemical sensors for measuring ozone have not been shown to be as robust.

Specific comments:

P.1, l. 30 "organic decomposition"?  Some biogenic VOCs are emitted through decomposition processes, but other natural sources like isoprene, terpenes, and some alcohols are emitted directly from plants.

We have edited the manuscript to just say "biogenic processes" as organic decomposition is a sub-section of processes by which there are VOC biogenic sources.

P.2, l. 42-44 There is nothing in the Beekman et al., 1997 reference about tethered balloons over water (it does discuss tropopause folding events).  Is there supposed to be a different reference for the first part of this sentence?  But really, the two parts of this sentence don't go together (ground to 1500 m vs. upper troposphere).

That reference was incorrect. The references have been updated and the sentence edited.

l. 48 I think this reference should be to Li et al., 2020 (comparison with the airship), not 2021 (primarily VOCs, and I saw no mention of an airship in the manuscript). Is Li et al., 2020 the most closely related paper to this manuscript (or perhaps that is Guimaras et al., 2020, or Gronoff et al., or several of them)? It does use a fixed-wing UAS rather than a hexacopter though.  But it seems to have a thorough evaluation section of the instruments and measurements.  It seems like the discussion section of this manuscript might need to include a bit more related to this paper.  Are your results comparable or similar to Figure 5a (or 7a, or 8) in Li et al., 2020?  In addition, please take a look at papers citing Li et al., 2020.  A few relevant ones are cited here (such as Q. Chen et al., 2020), but I think there are a couple of others that might be cited as well.  How about L. Chen et al., 2022?  I did not do a thorough search; the authors should do that.

You are correct, the reference should be Li et al 2020. We have looked into more references that the reviewer suggests. Wu 202, Chen 2019, Chem 2022 are all articles which can be described in a revised manuscript.

l.49 What is the correct reference here?

This should be Li et al 2018 ("Three-dimensional analysis of ozone and PM2.5 distributions obtained by observations of tethered balloon and unmanned aerial vehicle in Shanghai, China" *Stochastic Environmental Research and Risk Assessment*) instead of Li 2020. I did not notice anything in either Li et al. paper about Generalized Additive Models, but I did not read either of them thoroughly.

P.3, l.82 That is great that there is "improved performance and viability" but is that shown or demonstrated in the following sections? How can you do that without referring back to the results in the cited literature?

This sentence refers to the improvements to performance between the Park Falls, WI experiment and the Lake Michigan shoreline experiments outlined in this manuscript. This sentence has been edited for clarity in the revised manuscript.

P.5, l. 125 A 15 minute flight time is not ideal. Is there any way to get a similar platform with longer flight duration? (Again, this can be addressed in the discussion section.)

Yes, we are able to accomplish longer flights with different UAS (namely in experiments conducted in 2021 and 2022 with DJI M300 UAS). As this is referring to the experiments that occurred in 2020, we have added comments to the discussion with regards to improvements which could be made.

P.6, l. 180 Why does the filter need batteries or power? Perhaps I don't understand what the filter is, or what it is used for.

The filter does not need batteries. The sentence has been modified for clarity.

P.7, l. 184 Are these the actual accuracy and precision (considering the comparisons with other instruments) or just calculated from the formula from 2B? line 202 would suggest that the accuracy is not as good in flight. And compare with l. 245-246 and l. 252. Seems like the text needs to be made consistent on this.

Line 184 is referring to the calculation from 2B Tech and the rest of the analysis in the paper is to test the accuracy of the instrument in flight against a) tower observations or b) ground observations made at a similar inlet height to a hovering altitude for the UAS. A revised manuscript has addressed the clarity in line 184.

P.8, Table 1 The gradients measured by the POM were generally not distinguishable from zero.  So the statement on l. 201 is technically true, but not very helpful.  Glad to see that the results led to the subsequent improvements described later in that paragraph.

We agree that the comparison with tower observations are not great. The goal of sharing this table is to address the discrepancy with the absolute ozone measurements, the high noise of the observations and the understanding that if the gradients were closer to correct, accuracy could be improved by correcting for a zero-offset.

P.11, Figure 3a How does this figure compare with a similar one in Li et al. 2020?  (See earlier comments above.)  Again, this can be addressed in the discussion section or wherever it makes the most sense.

This figure has some similarities for the Li et al 2020 figure 5a, where they saw a linear fit of $0.7x - 7$ for a POM correlation to a regulatory ozone measurement instrument standard. The difference between our measurement and theirs is that we see more observations along the 1:1 line with higher ozone concentrations deviating the most from the center line, whereas the Li et al 2020 paper showed a consistent linear response at ~70% of the regulatory O3 measurement. Language about this comparison has been added to the revised manuscript.

P.14, l. 313 This sentence is a little confusing, with both tethered balloons and UAS.  I think it can be changed slightly to make it clearer.

So modified to: "Some researchers have successfully used UAS for vertical ozone profiles up to 1000 m using tethered balloons (Li et al., 2020) and a UAS-mounted thermally-insulated POM in the winter (Chen et al. 2020) "

P.15, Figure 4 I find it hard to distinguish the two profiles on June 18.  By adding a top axis for ozone, you would have 4 traces on panels b, c, and d, so that might be confusing too.  Perhaps just making the traces line+symbols (by adding reasonably thick gray and black lines for the two profiles, respectively, to the color-coded circles) it would be easy enough to follow.  Right now, I had to examine this figure very closely while reading the text on P. 13-14 in order to understand it.

As per the reviewer 1 comments, the panel can be made differently to make the AM and PM flights more distinguishable. A new figure will replace this one in the revised manuscript. Lines will be added to the figure in the revised manuscript.

P.16 After editing the rest of the paper, perhaps the conclusions section could be strengthened and made more useful to readers.

With the increased focus on the measurement techniques discussed in this paper, the conclusions have been edited to align with the manuscript revisions.

P.17-24 There is an extensive reference section, but a few of the references I checked do not seem to correspond to what is in the main text of the manuscript.  Is it possible to check at least the most important references against the text?  Maybe all of them?

Will do. The references will be corrected in the revised manuscript.

Figure S3 I can't tell the difference in the symbols between the two tower instruments.  But that's probably OK (if they agree with each other); the colors clearly mark the different elevations.  In the legend, can you put the two 122 m symbols next to each other?  The figure clearly shows the data from both the tower and the UAS.

Yes, this figure can be made more distinguishable.

I don't think you really need all the Figures S4-S10.  Just one or two for reference would be fine.

We'll keep Figure S8 as it is a nice demonstration of a lake breeze.

Perhaps the same comment for Figures S12-16, though these are at least related to the data shown in Figure 4.

OK, they have been removed from the final manuscript.

I definitely think that some of Figures S17-21 could be dropped.

OK, they have been removed from the final manuscript.

In Figure S22, are the dashed lines a running average?  Perhaps that should go into the caption.

 Yes. We have added a description to the legend.

Technical and proofreading comments:

P.2, l. 57 "create"?

So edited.

P.3, l. 93 "and"?  "on land"?

"And" was edited. Not sure where "on land" is in reference to.

P.5, l. 115 Are the times correct for 2020 flights?  Just wondering, because 6 pm is later than 11 am.  Maybe just reorder the two times.

So edited.

l. 130-132 This sentence is a little odd-sounding. I assume the UAS measurements were just a small part of the overall campaign. (It's fine up to "shoreline", but then rest of the sentence implies that the UAS was the purpose of the project.)

The campaign was just the UAS measurements with some additional ground observations (namely the wind-pro lidar).

P.6, l. 168 Please add a comma after "spectroscopy".

So edited.

P.13, l. 308 What do you mean by "fumigation"?  (This may be OK, I'm not sure.)

We mean vertical missing from pollutant emissions at the surface. We replaced fumigation with "transport" to simplify the statement.

---

## Author Comment (AC2)

R1: amt-2023-143

We thank the reviewer for their thorough review of the manuscript. Author's comments are in blue.

Reviewer comments for "Observing Low Altitude Features in Ozone Concentrations in a Shoreline Environment via Unmanned Aerial Systems" Josie K. Radtke et al. Atmos. Meas. Tech. amt-2023-143 Recommendation: Reject General Comments This paper presents a summary of uncrewed aircraft system (UAS) flights performed during the CHEESEHEAD19, PEcorINO, and WiscoDISCO20 campaigns that collected vertical profiles of atmospheric ozone measurements. The need for these types of measurements in general is well motivated, but the objectives of the paper are not well stated and in general the paper lacks discussion on the unique aspects of the sensor integration, sophisticated comparisons with reference measurements, and outlooks on future applications. While reasonably appropriate in scope for the journal Atmospheric Measurement Techniques, in my opinion there are issues with this study's presentation, experimental procedure, and scientific significance that would require substantial revisions before publication. Fatal Flaw In my opinion, this study is limited in regard to its overall contribution to the scientific literature, particularly in the realm of atmospheric observations with UAS. While the authors did a commendable job discussing the need for low-level observations of ozone in coastal environments, the primary results (a collection of vertical profiles to 120 m altitude with some comparisons to tower-based sensors) of the study are mostly proof-of-concept measurements collected with commercially available UAS airframes and sensors. Typically, these types of studies in AMT are focused on the design of custom-built UAS or unique sensor package integration (e.g., Altst¨adter et al., 2015; Segales et al., 2020; Hamilton et al., 2022), but that does not seem to be the focus of the present article. Otherwise, to my knowledge, there have already been a handful of studies collecting observations of atmospheric trace gases (including ozone) with UAS in a more systematic manner than the limited selection of cases presented here (e.g., Schuyler and Guzman, 2017; Schuyler et al., 2019; Krautwurst et al., 1 2021; Bretschneider et al., 2022, and references therein). Considering these factors, in my opinion the revisions necessary to improve this paper's contributions to the existing literature are too substantial at this time such that the submission should be rejected. However, I do believe the content of this study may warrant submission to a data journal such as Earth System Science Data to complement the data repositories cited at the end of the article.

It is heartening to hear that the reviewer finds this article "reasonably appropriate in scope for the journal Atmospheric Measurement Techniques" despite their feeling that this study contains what they deem as a fatal flaw. To speak to the state of where this manuscript lies within the context of other observations using UAS:

Alstadter et al (2015) describes PM2.5 observations on UAS, not $O_3$.

Seagales et al. (2022) describe thermos-hygrometer sensors on UAS, not $O_3$.

Hamilton et al, (2022) describes the DataHawk2 UAS as an atmospheric thermodynamic observation system for measurements of T, RH, P and wind speed and direction, not $O_3$.

The reviewer suggesting that a paper for AMT must require engineering of a new platform instead of careful study of the appropriate measurement strategy for combining commercially-available systems requirement for publishing a technique in this journal. The aim and scope of the journal, as stated, is "The main subject areas comprise the development, intercomparison, and validation of measurement

instruments and techniques of data processing and information retrieval for gases, aerosols, and clouds. Papers submitted to AMT must contain atmospheric measurements, laboratory measurements relevant for atmospheric science, and/or theoretical calculations of measurements simulations with detailed error analysis including instrument simulations." The papers must contain atmospheric measurements, which we present in this manuscript. Ultimately, we hope that this experiment aligns with the scope by describing a method for investigating vertical ozone profiles in the atmosphere using UAS backed up by evaluation of precision and accuracy of those observations.

We argue that this manuscript presents an analysis of intercomparison of UAS platform measurements with ground- or tower- based measurements which indicate pitfalls of putting a POM on any commercially-available UAS (as shown from the results of the tower comparisons during CHEESEHEAD) and the improved flight and calibration parameters which lead to improved accuracy and precision (During WiscoDISCO2020).

The articles in which the reviewer reflects on the other measurements:

Schuyler, T. J., S. C. C. Bailey, and M. I. Guzman (2019) measured $CO_2$ $CH_4$ and $NH_3$.

Schuyler, T. J., and M. I. Guzman (2017) propose techniques for measuring $CO_2$, $CH_4$ and $NH_3$ on UAS. This paper is not a comprehensive look at measurements of $O_3$ on UAS.

Krautwurst, S., and Coauthors, 2021: Quantification of CH4 coal mining emissions in Upper Silesia by passive airborne remote sensing observations with the Methane Airborne MAPper (MAMAP) instrument during the CO2 and Methane (CoMet) campaign. Atmospheric Chemistry and Physics, 21 (23), 17 345–17 371, https://doi.org/ 10.5194/acp-21-17345-2021.

    Looked at $CO_2$ and $CH_4$ by remote sensing.

Bretschneider, L., and Coauthors, 2022: MesSBAR—Multicopter and Instrumentation for Air Quality Research. Atmosphere, 13 (4), 629, https://doi.org/10.3390/atmos13040629.

    This manuscript does go over the measurement of $O_3$ with a POM and with AlphaSense electrochemical sensors. However, this manuscript does not address issues of the POM measurement accuracy in comparison to ground observations.

*Major Comments*

1. While the motivation for the study is reasonably established, it is not immediately clear in the introduction what the objectives of this study are outside of generally assessing some UAS vertical profile measurements of ozone. At the end of the introduction, please explicitly outline the relevant scientific questions, hypotheses, and/or novel concepts this paper will present.

The goal of WiscoDISCO2020 was to investigate the vertical profiles of ozone at a shoreline location impacted by high ozone episodes. The lake breeze phenomenon at that specific Location in Chiwaukee Priarie WI hosts a regulatory site at a shoreline state natural area, which is one of the few in Wisconsin which regularly exceed federal ozone standards. The large sources of emissions for ozone precursors are mainly concentrated in the Chicago metro area and the presence of Lake Michigan provides an inverted atmosphere at times in which to trap said pollutants. The role of the inversion over Lake Michigan, the advection of pollutants over Lake Michigan and then back on land

during the meso-scale meteorological phenomenon of the Lake Breeze is the focus of the WiscoDISCo field campaigns. This manuscript firstly outlines how the instrumentation was tested in a non-lake shore environment (during CHEESEHEAD19) and improvements to the experiment improved instrumentation performance for the first WiscoDISCO field campaign in 2020.

2. Although not the primary focus of the study, I am rather concerned with the quality of meteorological observations collected by the iMet sensors due to their siting onboard the UAS airframes. In particular, there have been numerous investigations on the placement of temperature sensors to mitigate the influences of solar radiation, heat from the UAS motors, and heat from the body of the UAS itself while still maintaining adequate ventilation (see the discussions in Greene et al., 2018, 2019; Barbieri et al., 2019; Islam et al., 2019; Kimball et al., 2020). While the iMet-XQ2 mounting onboard the Typhoon H (Figure S1) seems reasonably well sited, the position on the MJ600 (Figure S2) likely resulted in biases due to lack of ventilation and exposure to direct sunlight and heat from the black aircraft body. While this is not something that can be corrected for necessarily, please at least include a discussion on this in the results.

   Our results from comparisons with other monitors show that the performance on the Typhoon UAS was reasonable but there was improved performance on the DJI M600 (see Figure 3b). A discussion of the relevant studies on iMET-XQ2 and the observed performance during this study has been added to the paper.

3. In Sections 3.1 and 3.2, UAS measurements are compared with tower-based and groundbased references. However, this is mostly presented as single cases summarized in tables 1 and 2 as well as figure 3. For a calibration procedure, I think a more thorough analysis is warranted, especially to contextualize the cases presented in section 3.3. For example, how many individual data points were collected on each day and at each level?
   A table with flight numbers, time windows and number of data points per instrument or platform has been added to the SI to outline the data presented in Section 3.3. The finalized dataset in Zenodo is discoverable which contains the data averaged to the 5-minute hovering peroids.
   Additionally, bulk statistics across all days such as the mean, median, and standard deviation of the differences between UAS and tower measurements of each variable would be pertinent. What conditions are present where the largest biases are observed?
   So if this is with regards to tower comparisons in section 3.1, an additional table has been added to address mean, median and standard deviations from the UAS -tower observations as a companion to Figure S3 in the SI in the revised manuscript.

4. The entire paper is building up towards the results from the cases in Section 3.3, with a lot of emphasis on the vertical distribution of ozone and temperature versus height across multiple days. These results are provided in Figure 4, with the discussion focusing on features of these vertical profiles. The current presentation and layout of this figure, however, make it difficult to follow the discussion in Section 3.3. For example, with emphasis on changes in the vertical, I recommend changing the layout of the subpanels to be organized horizontally instead of being stacked vertically so that the subpanels are taller than they are wide (or at least with an aspect ratio of 1:1). Additionally, it is difficult to tell the difference between the AM and PM profiles for 2 the June 16, 17, and 18 cases; please consider using different shapes (circles, squares, crosses, etc.) for the different profiles on the same day. I also strongly urge the use of a colorblind

friendly color palette that is also uniformly perceptive in place of the current rainbow color bar(see Stauffer et al., 2015).

The suggestion of the reviewer here is a good one, and an improved figure will be added to a revised manuscript which addresses this comment.

Finally, why was the choice made to use the HRRR PBL height as a reference in this figure? You mention there was a Doppler lidar present for the WiscoDISCO20 campaign, was this capable of producing PBL height estimates more locally? Otherwise, consider omitting the earlier discussions on the instruments not used for this current study.

HRRR PBL height is a metric which addresses how photochemical models are treating vertical profiles when computing photochemical ozone production. The use of the HRRR PBL height highlights the sub-grid scale of the vertical profiling. Also, the Doppler lidar instrument has a dead zone at low altitudes (<100 m AGL) in which no observations are made. The PBL heights at this location specifically lie within that dead zone during lake breeze times, so the vertical profile measurements and HRRR PBL height outputs help to highlight the scale of these lake breeze phenomena (not observable by Lidar to low altitudes). As per the response to Reviewer 2, much of the lidar discussion has been removed from the SI. Some comments about the utility of the HRRR PBL height have been added to a revised manuscript.

*Minor and Technical Comments*

1.  L26: Please remove the period at the start of the line.
    Done.
2.  L40: Please define the acronym UAS.
    Done.
3.  L42: Please remove the extra period between the citations and the start of the next sentence.
    Done
4.  L57: Spelling error: should "crate" read as "create"?
    Done.
5.  L86: Please define the acronym "UW" in UW-Eau Claire
    Done.
6.  L93: Spelling error: remove the "F" at the start of the word "and".
    Done.
7.  Section 2.2: Here I have a handful of suggestions for breaking this long first paragraph up into logical sections. First, at L138, the stentence starting "The main goal of this campaign..." could start a new paragraph. Similarly, break a new paragraph as L147 starting with "During WiscoDISCO20 UAS...".
    Done and Done.
8.  148: Please define the acronym "DNR."

    Done.

9.  L148–154: Did you use these instruments specifically in this study? Consider omitting this portion (see major comment 4).
    As per the Reviewer 2 comments, with removing discussion of the Doppler Lidar, we will omit comment on the Pandara and Doppler Lidar instrumentation.

10. L165: This would be a good place to highlight the total number of flights conducted in each period.

    A table for flights and flight times has been added to the SI.

11. L177: Add a space between "electrochemical sensors" and the following parenthetical citation.

    Done.

12. L194: Should this read "an intercomparison…" instead of just "n"?

    Yes. So changed.

13. L202–204: The sentence beginning "The UAS gradient observations…" is a bit hard to follow, please consider rewording.

    In combination of reviewer 1 and 2 comments, the statements here are being revised in the final document to the following:

    "Technically the overall comparison between tower gradients and UAS gradients show agreement; however the considerable uncertainties make both indistinguishable from zero (See Table 1). This evaluation demonstrated a likely source of inaccuracy with POM ozone observations, with significant offset from the absolute tower observations."

14. L208: I recommend breaking a new paragraph starting with the sentence "Improvements to the UAS sensor package…" 3

    Done.

15. Table 1 and surrounding discussion: This is perhaps semantic, but these are not necessarily gradients but rather just differences. Please consider changing the wording throughout, or computing the gradients by dividing the differences by the height between the sensors.

    Ok. Changing all language over to 'differences' may make the entire paper more difficult to distinguish what an observed vertical distribution was per platform and a comparison between two instruments. This can be addressed in a revised manuscript.

16. Table 2: Please include the total number of flights and/or individual samples that go into each mean and standard deviation presented here (see also major point 3). 4

    Ns have been added to the table.

*Reviewer 1 References*

Altst¨adter, B., and Coauthors, 2015: ALADINA – an unmanned research aircraft for observing vertical and horizontal distributions of ultrafine particles within the atmospheric boundary layer. Atmospheric Measurement Techniques, 8 (4), 1627–1639, https://doi.org/ 10.5194/amt-8-1627-2015.

Barbieri, L., and Coauthors, 2019: Intercomparison of Small Unmanned Aircraft System (sUAS) Measurements for Atmospheric Science during the LAPSE-RATE Campaign. Sensors, 19 (9), 2179, https://doi.org/10.3390/s19092179.

Bretschneider, L., and Coauthors, 2022: MesSBAR—Multicopter and Instrumentation for Air Quality Research. Atmosphere, 13 (4), 629, https://doi.org/10.3390/atmos13040629.

Greene, B. R., A. R. Segales, T. M. Bell, E. A. Pillar-Little, and P. B. Chilson, 2019: Environmental and Sensor Integration Influences on Temperature Measurements by Rotary-Wing Unmanned Aircraft Systems. Sensors, 19 (6), 1470, https://doi.org/10.3390/s19061470.

Greene, B. R., A. R. Segales, S. Waugh, S. Duthoit, and P. B. Chilson, 2018: Considerations for temperature sensor placement on rotary-wing unmanned aircraft systems. Atmospheric Measurement Techniques, 11 (10), 5519–5530, https://doi.org/10.5194/ amt-11-5519-2018.

Hamilton, J., G. de Boer, A. Doddi, and D. A. Lawrence, 2022: The DataHawk2 uncrewed aircraft system for atmospheric research. Atmospheric Measurement Techniques, 15 (22), 6789–6806, https://doi.org/10.5194/amt-15-6789-2022.

Islam, A., A. L. Houston, A. Shankar, and C. Detweiler, 2019: Design and Evaluation of Sensor Housing for Boundary Layer Profiling Using Multirotors. Sensors, 19 (11), 2481, https://doi.org/10.3390/s19112481.

Kimball, S. K., C. J. Montalvo, and M. S. Mulekar, 2020: Assessing iMET-XQ Performance and Optimal Placement on a Small Off-the-Shelf, Rotary-Wing UAV, as a Function of Atmospheric Conditions. Atmosphere, 11 (6), 660, https://doi.org/10.3390/atmos11060660.

Krautwurst, S., and Coauthors, 2021: Quantification of CH4 coal mining emissions in Upper Silesia by passive airborne remote sensing observations with the Methane Airborne MAPper (MAMAP) instrument during the CO2 and Methane (CoMet) campaign. Atmospheric Chemistry and Physics, 21 (23), 17 345–17 371, https://doi.org/ 10.5194/acp-21-17345-2021.

Schuyler, T. J., S. C. C. Bailey, and M. I. Guzman, 2019: Monitoring Tropospheric Gases with Small Unmanned Aerial Systems (sUAS) during the Second CLOUDMAP Flight Campaign. Atmosphere, 10 (8), 434, https://doi.org/10.3390/atmos10080434. Schuyler, T. J., and M. I. Guzman, 2017: Unmanned Aerial Systems for Monitoring Trace Tropospheric Gases. Atmosphere, 8 (10), 206, https://doi.org/10.3390/atmos8100206. 5 Segales, A. R., B. R. Greene, T. M. Bell, W. Doyle, J. J. Martin, E. A. Pillar-Little, and P. B. Chilson, 2020: The CopterSonde: An insight into the development of a smart unmanned aircraft system for atmospheric boundary layer research. Atmospheric Measurement Techniques, 13 (5), 2833–2848, https://doi.org/10.5194/amt-13-2833-2020. Stauffer, R., G. J. Mayr, M. Dabernig, and A. Zeileis, 2015: Somewhere Over the Rainbow: How to Make Effective Use of Colors in Meteorological Visualizations. Bulletin of the American Meteorological Society, 96 (2), 203–216, https://doi.org/10.1175/ BAMS-D-13-00155.1.

---

## Referee Report (RR1)

**Reviewer comments for "Observing Low Altitude Features in Ozone Concentrations in a Shoreline Environment via Unmanned Aerial Systems"**

Josie K. Radtke et al.
*Atmos. Meas. Tech.*

amt-2023-143

**Recommendation: Major Revisions**

**General Comments**

I appreciate the work by the authors towards addressing the reviewer comments in this updated manuscript draft, and overall I am more satisfied with the quality of this study. Most of my specific concerns about measurement quality were commented on in the updated draft, and the relevance of this paper within the literature is established much more clearly. After addressing a handful more comments I am willing to consider this paper for publication.

**Major Comments**

1. Figure 4: Thank you for updating the layout of this figure for clarity. As per my original comment, I additionally request the authors to update the color palette on this figure to something other than rainbow, as I find it difficult to read differences in adjacent points without a perceptually uniform palette. Depending on the programming language used to make this figure, I recommend a color map from the "cmocean" package (links for MATLAB, Python).

**Minor and Technical Comments**

1. For a paper focusing on novel observations of $O_3$, I would be in favor of moving the supplemental figures S1 and S2 into the main paper for easier reference, especially if they are referenced in the text anyways. Please consider moving them to Section 2.

2. P8, L210: Thank you for adding this text with discussion on the iMet-XQ2 performance. There is a double reference for Kimball et al. at the beginning of this line, please update.

---

## Author Response (AR2)

Thank you to the thoughtful reviewers of this manuscript. Their input has helped improve the quality of the manuscript.

Author comments are in blue.

**Review 1**

**General Comments** I appreciate the work by the authors towards addressing the reviewer comments in this updated manuscript draft, and overall I am more satisfied with the quality of this study. Most of my specific concerns about measurement quality were commented on in the updated draft, and the relevance of this paper within the literature is established much more clearly. After addressing a handful more comments I am willing to consider this paper for publication.

Thank you for your careful consideration of this manuscript.

**Major Comments 1**. Figure 4: Thank you for updating the layout of this figure for clarity. As per my original comment, I additionally request the authors to update the color palette on this figure to something other than rainbow, as I find it difficult to read differences in adjacent points without a perceptually uniform palette. Depending on the programming language used to make this figure, I recommend a color map from the "cmocean" package (links for MATLAB, Python).

This figure has now been remade with a different colormap.

**Minor and Technical Comments 1**. For a paper focusing on novel observations of O3, I would be in favor of moving the supplemental figures S1 and S2 into the main paper for easier reference, especially if they are referenced in the text anyways. Please consider moving them to Section 2. 2.

Both images have been added as Figure 2. The rest of the figures have been renumbered accordingly.

P8, L210: Thank you for adding this text with discussion on the iMet-XQ2 perfor⬚mance. There is a double reference for Kimball et al. at the beginning of this line, please update.

The second reference to Kimball has been deleted.

**Review 2:**

This paper has been rewritten to address most of the comments of the reviewers. It is certainly improved from the first version, and is now more limited by the experimental work rather than by the writing and organization. To reiterate from my previous review, the POM is an attractive choice for balloons (because it can be used on small balloons without obtaining special flight permissions) and UAS (for the same reason, and for its light weight and small size). **But is it really adequate for atmospheric measurements? Are the improvements and challenges needed to make it work mostly related to properly integrating it into the right UAS, or do improvements (or modifications by the user) need to be made to the sensor itself? The authors may not feel comfortable making these statements, but the answers to these questions would be useful to the community.** I would be happy with "suggestions for further improvements" being added to the Conclusions section, although the second-to-last paragraph on p. 15 (discussion of Li et al., 2020) may be a better spot, given that the authors have started to address this in that section. It is also OK

for the community to simply see what has been done and draw their own conclusions - I don't want to force something that is not fully supported by the work.

We agree that a section that summarizes what we learned from this experiment and what suggestions a reader should take home for implementation or further improvements is worthwhile to include. We added a suggestions section to Conclusions:

"In this study, the POM performance on UAS was improved by inlet positioning and slow flight parameters, top-mount placement on a robust UAS and increasing the rate of calibrations to pair each calibration with specific battery power source improved the precision and accuracy. However, added thermal insulation, as described by Li et al, appears another promising additional consideration for improved performance of the POM on UAS. The POM appears to be a robust enough instrument for course atmospheric measurements in the atmosphere (to 2 ppb precision) but integration onto a UAS should be carefully considered.

Specific comments (line #s from the "Track changes" version, not the final version):

P.2, l. 35 Certainly add "e.g." before Kaser et al., perhaps elsewhere as well if appropriate.

Added to line 35

l. 43-45 Again, there is nothing in the Beekman et al., 1997 reference about tethered balloons over water. My confusion about this sentence is that, as written, it looks like it is one subject or thought, but actually I guess it is three different things. If you add "and" before "associating high ozone" it will make much more sense. If the large set of references at the end of the sentence are split up so some follow "profiles over water/urban", some follow "ground to 1500 m", and some follow "UT/tropopause folds" it will be helpful.

This sentence has been recrafted to specifically identify some aspects of each of the sources cited.

It now reads:

Tethered balloons have been used to study vertical ozone (Demuer et al., 1997; Peng et al., 2008; Knapp et al., 1998; Zhang et al., 2019; Greenberg et al., 2009), and meteorological conditions (Chandrasekar et al., 2003) gathering data at heights ranging from ground level to 1500 meters above ground level, which included evaluations of episodes of biomass burning (Xu et al., 2018) and mesoscale modeling of ozone in the upper troposphere (Peng et al., 2008).

l. 50 "lower free troposphere"?

now says "in the lower troposphere"

P.6, l. 145 "ozone concentrations" or "mixing ratios" or just "ozone" instead of "measurements".

Deleted concentrations.

l. 172 "install" or "add" instead of "put"?

replaced "put" with "place"

l. 184-85 It seems that these two sentences could be combined into one, "The 2B Tech personal ozone monitor, POM, measures atmospheric ozone concentrations via UV absorption…" Also, I don't think this paragraph is an accurate summary of Wilson and Birks, 2006, in that the artifact can affect both dual and single cell instruments. It would be best if the authors simply state how they addressed the issue of artifacts from (changing) humidity, either using something provided by 2B Technologies, or their own design, or if they did nothing. Fine to have a short, accurate sentence about what causes the artifacts, from Wilson and Birks and/or subsequent work.

The sentences have been rearranged to stress how the POM works following the reviewer's suggestion.

P.9, l. 233 "both indistinguishable from zero". In table 1, 3 out of 4 tower gradient measurements are statistically different from zero. Not sure what "both" means here. And I'm not sure why the gradient is more important than the actual values measured. They may look better in comparison, but they have twice the uncertainty. It does suggest whether the reason for disagreement is a zero offset, or something else.

Many boundary-layer parameterizations for mixing, flux-profile relationships, and so on are functions of gradients of scalars, momentum, or heat more so than absolute values. For some applications that may be more important, while for others the absolute magnitudes matter. We have updated the sentence to clarify meaning.

l.237 "larger differences" – larger than what? A simple rewrite of this sentence or section should be able to fix this.

Removed the word larger.

Comparing Table 1 to Figure S3, for July 16 the POM UAS gradient looks like it should be close to -20 ppb. Or is there a second blue square very close to the tower data?

There are overlying datapoints. The figure (Now S1) has been edited to only have one data point per altitude which was used to calculate the ozone gradient from the POM observations.

Actually, some of the ToF data in S3 are a little suspicious too, particularly where they go close to zero on July 11. There are a number of outliers in the tower data, and they (very nearly) all seem to be triangles, or ToF measurements. If the averaging for both tower instruments are done the same way, it suggests that there are some things to be cleaned up in the ToF data. This is beyond the scope of this paper, but affects how much one might trust the comparison. In this case, the results clearly indicate that some improvements needed to be made in the O3/UAS system, as discussed

next.
Table 2 –

We have updated Figure S1 (in previous version S3) to include the quality controlled TOF data that was posted to the CHEESEHEAD repository. These data do not include outliers that are above 75 ppb. The lower values from the TOF are considered indicative of reactive chemistry with biogenic VOCs or soil NOx within the canopy (see Vermeuel et al 2021 GRL fluxes) or, less routinely, from reactions with NOx emitted from a station power generator that was irregularly tested. It is not uncommon for O3 values to reach <10 ppb at night in this region, as we have recorded values as low as 4 ppb at this same site in Fall 2020 (Vermeuel, et al. 2023) with the same Thermo 49i instrument used in this study. Further, the 1s-averaged limit of detection for this ToF is ~10 ppt (Novak et al., 2020), which allows for quantification at these low [O3] levels.

Ozone deltas were recalculated for Table 1, based on the updated dataset from the CHEESEHEAD19 repository which did not include data from July 16, so that data point was omitted from the table, Table S1 was also updated.

Sources:

Vermeuel, M. P., Cleary, P. A., Desai, A. R., & Bertram, T. H. (2021). Simultaneous measurements of O3 and HCOOH vertical fluxes indicate rapid in-canopy terpene chemistry enhances O3 removal over mixed temperate forests. Geophysical Research Letters, 48(3), e2020GL090996.

Vermeuel, M. P., Novak, G. A., Kilgour, D. B., Claflin, M. S., Lerner, B. M., Trowbridge, A. M., ... & Bertram, T. H. (2023). Observations of biogenic volatile organic compounds over a mixed temperate forest during the summer to autumn transition. Atmospheric Chemistry and Physics, 23(7), 4123-4148.

Novak, G. A., Vermeuel, M. P., & Bertram, T. H. (2020). Simultaneous detection of ozone and nitrogen dioxide by oxygen anion chemical ionization mass spectrometry: a fast-time-response sensor suitable for eddy covariance measurements. Atmospheric Measurement Techniques, 13(4), 1887-1907."

For the measurements at Park Falls in 2020, why not use the DJI hexacopter?

We did not have access to the DJI hexacopter in September 2020 for the measurements in Park Falls as it was owned by Purdue University.

The table is fine for comparing the iMet measurements, but the UAS had already been upgraded.

I don't quite understand this comment. For all observations at the Tower site, the Yuneec Typhoon H UAS was used. For the Summer 2020 Chiwaukee Prairie observations, the DJI M300 was used in the collaborative project with Purdue University.

P.10, l. 260-61 I'm not sure why the word "dimensionality" is used. The atmosphere is inherently 3D, and any experiment or analysis needs to consider that. (Or it is 4-D including time, and the short duration of small UAS flights is not necessarily well-matched.) Maybe it is the small-scale vertical (and perhaps horizontal) structures that are well matched for UAS. Sorry to be so picky here, I think I know what you mean, but I also think this can be improved.

So edited to state

"The viability for UAS-mounted ozone observations to capture low-altitude features in ozone is well-matched to the **small-scale vertical structure** of marine layer ozone concentrations in a near-shore environment."

P.15, l. 352-370 Thank you for including this! Worthwhile to check wording in places though – for example, "did a heating method" on l. 360. Maybe something like "Li et al. state only that the regulatory monitor for comparison used a heating method for removing water vapor interference, instead of…" (It's hard to avoid overusing the verb "use", but "employ" is another option.)

So changed to:

Li et al. address only that the regulatory monitor they used for comparison which employed an in-line heating method for removing water vapor interference, instead of a dual-cell active subtraction in parallel as is typical for other regulatory monitors.

See sentence on l. 373 – besides "used" and "using", it has "uncrewed aerial systems" as a type of "UAS". Could change that to the type of system – hexacopter, etc.? Also, I'm not sure if there is a "water vapor absorption" or an interference; please check that carefully and other similar mentions in the paper.

Edited this section for clarity

l. 368 "constant ascents"?

Edited

P.16, l. 378 "Lake Michigan"

Edited

P.17 Figure 4 is much easier to read now, despite (or perhaps partially because of) the shorter horizontal scale for potential temperature.
P.18, l. 390 Can you make this first sentence of the Conclusions more related to this paper, instead of the general "has a proven utility", which could be determined from the existing literature?

l. 391 "including" instead of "included" or rewrite this sentence. (The content is fine.

So edited

l. 395 "towers"?

Not sure what this is in reference to

Line 33 on p2 is the only place where the word "towers" shows up.

References:

Chandrasekar, A., Philbrick, C. R., Doddridge, B., Clark, R., and Georgopoulos, P.: A comparison study of RAMS simulations with aircraft, wind profiler, lidar, tethered balloon and RASS data over Philadelphia during a 1999 summer episode, Atmospheric Environment, 37, 4973-4984, 10.1016/j.atmosenv.2003.08.030, 2003.
DeMuer, D., Heylen, R., VanLoey, M., and DeSadelaer, G.: Photochemical ozone production in the convective mixed layer, studied with a tethered balloon sounding system, Journal of Geophysical Research-Atmospheres, 102, 15933-15947, 10.1029/97jd01211, 1997.
Greenberg, J. R., Guenther, A. B., and Turnipseed, A.: Tethered balloon-based soundings of ozone, aerosols, and solar radiation near Mexico City during MIRAGE-MEX, Atmospheric Environment, 43, 2672-2677, 10.1016/j.atmosenv.2009.02.019, 2009.
Knapp, K. G., Jensen, M. L., Balsley, B. B., Bognar, J. A., Oltmans, S. J., Smith, T. W., and Birks, J. W.: Vertical profiling using a complementary kite and tethered balloon platform at Ferryland Downs, Newfoundland, Canada: Observation of a dry, ozone-rich plume in the free troposphere, Journal of Geophysical Research-Atmospheres, 103, 13389-13397, 10.1029/97jd01831, 1998.
Peng, Y. P., Chen, K. S., Lou, J. C., Hwang, S. W., Wang, W. C., Lai, C. H., and Tsai, M. Y.: Measurements and Mesoscale Modeling of Autumnal Vertical Ozone Profiles in Southern Taiwan, Terrestrial Atmospheric and Oceanic Sciences, 19, 505-514, 10.3319/tao.2008.19.5.505(a), 2008.
Xu, Z. N., Huang, X., Nie, W., Shen, Y. C., Zheng, L. F., Xie, Y. N., Wang, T. Y., Ding, K., Liu, L. X., Zhou, D. R., Qi, X. M., and Ding, A. J.: Impact of Biomass Burning and Vertical Mixing of Residual-Layer Aged Plumes on Ozone in the Yangtze River Delta, China: A Tethered-Balloon Measurement and Modeling Study of a Multiday Ozone Episode, Journal of Geophysical Research-Atmospheres, 123, 11786-11803, 10.1029/2018jd028994, 2018.
Zhang, K., Zhou, L., Fu, Q. Y., Yan, L., Bian, Q. G., Wang, D. F., and Xiu, G. L.: Vertical distribution of ozone over Shanghai during late spring: A balloon-borne observation, Atmospheric Environment, 208, 48-60, 10.1016/j.atmosenv.2019.03.011, 2019.